# Energetics of the microsporidian polar tube invasion machinery

Ray Chang[1], Ari Davydov[2], Pattana Jaroenlak[2†], Breane Budaitis[2], Damian C Ekiert[2,3], Gira Bhabha[2]*, Manu Prakash[1,4]*

[1]Department of Bioengineering, Stanford University, Stanford, United States; [2]Department of Cell Biology, New York University School of Medicine, New York, United States; [3]Department of Microbiology, New York University School of Medicine, New York, United States; [4]Woods Institute for the Environment, Stanford University, Stanford, United States

**\*For correspondence:**
gira.bhabha@gmail.com (GB);
manup@stanford.edu (MP)

**Present address:** †Center of Excellence for Molecular Biologyand Genomics of Shrimp, Department of Biochemistry, Faculty of Science, Chulalongkorn University, Bangkok, Thailand

**Competing interest:** The authors declare that no competing interests exist.

**Abstract** Microsporidia are eukaryotic, obligate intracellular parasites that infect a wide range of hosts, leading to health and economic burdens worldwide. Microsporidia use an unusual invasion organelle called the polar tube (PT), which is ejected from a dormant spore at ultra-fast speeds, to infect host cells. The mechanics of PT ejection are impressive. *Anncaliia algerae* microsporidia spores (3–4 µm in size) shoot out a 100-nm-wide PT at a speed of 300 µm/s, creating a shear rate of 3000 s⁻¹. The infectious cargo, which contains two nuclei, is shot through this narrow tube for a distance of ~60–140 µm (Jaroenlak et al, 2020) and into the host cell. Considering the large hydraulic resistance in an extremely thin tube and the low-Reynolds-number nature of the process, it is not known how microsporidia can achieve this ultrafast event. In this study, we use Serial Block-Face Scanning Electron Microscopy to capture 3-dimensional snapshots of *A. algerae* spores in different states of the PT ejection process. Grounded in these data, we propose a theoretical framework starting with a systematic exploration of possible topological connectivity amongst organelles, and assess the energy requirements of the resulting models. We perform PT firing experiments in media of varying viscosity, and use the results to rank our proposed hypotheses based on their predicted energy requirement. We also present a possible mechanism for cargo translocation, and quantitatively compare our predictions to experimental observations. Our study provides a comprehensive biophysical analysis of the energy dissipation of microsporidian infection process and demonstrates the extreme limits of cellular hydraulics.

## eLife assessment

This **important** study combines experiments and fluid mechanics modeling to determine the mechanism of the ultrafast ejection of the polar tube of the Microsporidia parasite and of transport through this tube. The methods and the analysis, based on the variation of the viscosity of the external medium, are **compelling** and allow for the first time to discriminate among proposed ejection mechanisms. This approach where simple physical principles are used for distinguishing between mechanisms when the precise geometry is inaccessible through imaging is potentially applicable to other systems in microbiology.

## Introduction

### Microsporidia: opportunistic intracellular parasites

Microsporidia are single-celled intracellular parasites that can infect a wide range of animal hosts (*Keeling and Fast, 2002*). Microsporidia are most closely related to fungi, but diverged from other

species very early in the evolution of the fungal kingdom (*Capella-Gutiérrez et al., 2012*). In humans, microsporidia act as opportunistic pathogens, with the ability to infect several organ systems. Microsporidia infection in patients with compromised immune systems can be fatal (*Kotler and Orenstein, 1998*). Despite their medical importance, the treatment options for microsporidial diseases remain limited (*Han and Weiss, 2018*; *Maillard et al., 2021*). The prevalence of microsporidia is high; a systematic review in 2021 showed that the overall prevalence rate of microsporidia infection in humans was estimated to be 10.2%, and the contamination rate of water bodies with human-infecting microsporidia species is about 58.5% (*Ruan et al., 2021*). Infection of other animals, such as farmed fish, can lead to large economic burdens in countries that depend heavily on these industries (*Stentiford et al., 2016*). Current financial losses in Southeast-Asian shrimp farming alone are estimated to be on the order of billions of dollars each year (*Stentiford et al., 2016*). Microsporidia are not genetically tractable organisms at this time, which severely limits the study of their biology and infection process.

## Anatomy of a microsporidian spore

This study focuses on *Anncaliia algerae* (*Figure 1A*), a microsporidian species that can infect both humans and mosquitoes (*Weiss and Takvorian, 2021*). *A. algerae* spores can survive in ambient environments for months (*Becnel and Andreadis, 2014*). The protective microsporidian spore coat consists of 3 layers: (1) a proteinaceous exospore, (2) an endospore, of which chitin is the major component, and (3) a plasma membrane. Within the spore, the polar tube (PT) infection organelle is the most striking feature, visually appearing as a rib cage that surrounds other organelles. How spaces in distinct organelles are topologically connected within the spore remains ambiguous. It is likely that the PT is an extracellular organelle, which is topologically outside the plasma membrane, but inside the spore wall (*Cali et al., 2002*). The PT is anchored to the apical end of the spore via a structure called the anchoring disc, which presses up against the thinnest region of the endospore, and is the region from which PT firing is initiated. The PT is linear at the apical end of the spore, and then forms a series of coils, which terminate at the posterior end of the spore. The PT is arranged as a right-handed helix that interacts closely with other spore organelles, including a vacuole at the posterior end (known as 'posterior vacuole'), and a stack of membranes called the polaroplast at the anterior end. The posterior vacuole has been previously observed to expand during the germination process, and is thus thought to play a role during spore germination, potentially providing a driving force for translocating cargo through the PT (*Troemel and Becnel, 2015*). The polaroplast closely associates with the linear segment of the PT, and is thought to play a role in the initial stages of the germination process by swelling and exerting a force on the spore wall, causing it to rupture (*Keohane and Weiss, 1998*). It may also serve as a supplementary membrane source for the PT as it fires from the spore.

## Microsporidia eject the PT organelle at ultrafast speed to infect host cells

Microsporidian spores establish infection via a mechanism different from other parasites and pathogens (*Figure 1B–E*). The PT mediates invasion into a host cell via an ultra-fast physical process termed PT ejection (*Weidner, 1972*; *Schottelius et al., 2000*; *Franzen et al., 2005*). The PT, typically many times the length of the spore, is coiled up to fit inside a dormant spore. Once triggered, the spore rapidly shoots out the PT, which forms a conduit that transports the infectious cargo, or sporoplasm, into the host cell, in a process also known as germination (*Weidner, 1972*; *Schottelius et al., 2000*; *Franzen et al., 2005*). The PT of *A. algerae* is about 100-μm-long and only 100-nm-wide (*Jaroenlak et al., 2020*). Spores are capable of shooting the PT at a peak velocity up to 100–300 μm/s (*Frixione et al., 1992*; *Jaroenlak et al., 2020*, *Figure 1E*). Once fired, the extruded PT is roughly two times longer than when it is coiled in the dormant spore (*Jaroenlak et al., 2020*). Considering the thin cross-section of the tube (100 nm), the shear rate (defined as shear per unit time) experienced by the PT is on the order of 3000 s$^{-1}$, which is an order of magnitude larger than the wall shear rate on the human aorta (300–800 s$^{-1}$) (*Gogia and Neelamegham, 2015*). While the exact nature of the cargo being transported through the tube into the host is not known, it is thought that the entire contents of the microsporidian cell are likely to be transported. For *A. algerae*, this includes two identical nuclei and other organelles. Using these nuclei as a marker, translocation of cargo through the PT has recently

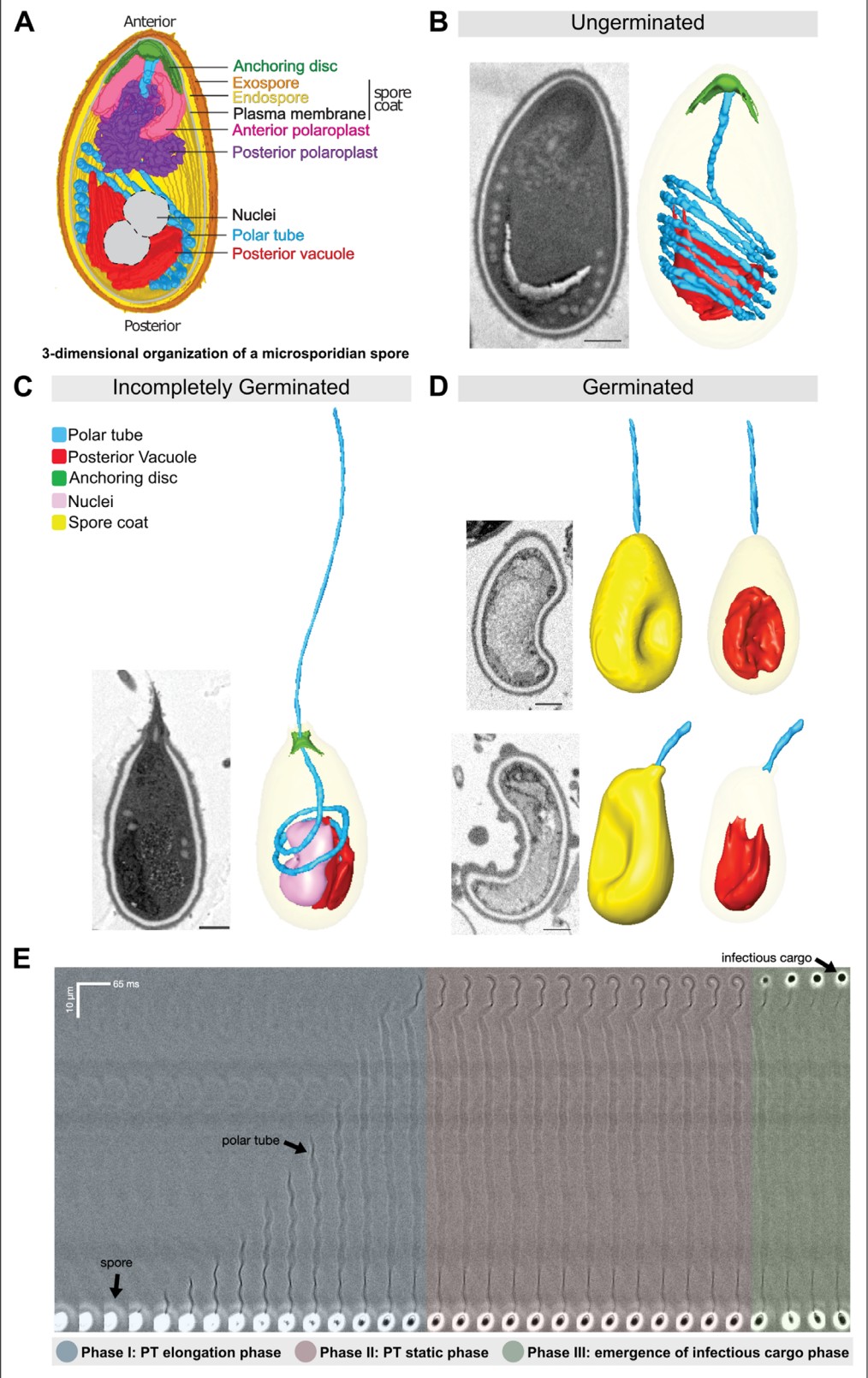

**Figure 1.** Morphology of germinating *A.algerae* spores. (**A**) Overall organization of organelles in an *A. algerae* spore. The spore coat consists of 3 layers: a proteinaceous exospore (orange), a chitin-containing endospore (yellow), and a plasma membrane. Within the spore, the polar tube (PT) (blue), which is the infection organelle, surrounds other organelles like a rib cage. The PT is anchored to the apical end of the spore via a structure called

*Figure 1 continued*

the anchoring disc (green). At the apical end, the PT is linear, and then forms a series of coils, which end at the posterior end of the spore. The PT interacts closely with other spore organelles, including the posterior vacuole (red), and a membranous organelle called the polaroplast (purple). The organization of the spore shown here comes from SBF-SEM data (bright colors) and TEM images (nuclei positioning, and plasma membrane, grey). (**B–D**) Examples of slices from SBF-SEM imaging and the corresponding 3D reconstructions for ungerminated (**B**), incompletely germinated (**C**) and germinated (**D**) *A. algerae* spores. Colored according to the color key shown in (**C**). All scale bars are 500 nm. (**E**) Kymograph of the PT ejection process in *A. algerae*. The PT ejection process can be divided into 3 phases: PT elongation phase (blue), PT static phase (pink), and emergence of infectious cargo phase (green). This kymograph was generated from data deposited in *Jaroenlak et al., 2020*.

been visualized by high-speed imaging (*Jaroenlak et al., 2020*), showing that cargo transport occurs on a timescale similar to PT extrusion.

## Lack of biophysical models explaining the microsporidian infection process

Because of the ultrafast nature of PT ejection and the high hydraulic resistance associated with an extremely thin tube (100 nm in diameter), historically it was thought to be impossible for infectious cargo to flow through the PT at a comparable speed to PT extension (*Ohshima, 1927*; *Weiser, 1947*; *Dissanaike and Canning, 1957*). Consequently, several hypotheses were proposed that were thought to be more physically plausible (see past reviews on this *Lom and Vavra, 1963*; *West, 1960*), and one of these hypotheses that gained popularity was termed 'jack-in-the-box' (*Ohshima, 1927*; *Weiser, 1947*; *Dissanaike and Canning, 1957*). In this hypothesis, the PT is proposed to rapidly spring out from the spore, with the infectious cargo attached to the end of the PT, thus getting sprung out at the same time (*West, 1960*). However, the jack-in-the-box model arises from observations in which external pressure was applied to spores, which may challenge the interpretation of the observations (*Dissanaike and Canning, 1957*; *West, 1960*).

Later experimental evidence, such as microscopic observations of PT extrusion (*Thomson, 1959*; *West, 1960*) and pulse-labeling of a half-ejected tube (*Weidner, 1982*), suggests that the PT ejection process is more likely a tube eversion process, in which the PT turns inside out as it is extruded, such that only the tip is moving during germination. As the PT extrudes, the infectious cargo squeezes through the PT and emerges at the other end. Although the eversion hypothesis is thought to be most likely, no quantitative biophysical analysis has been done on this process, leaving open the physical basis for the PT firing mechanism. Furthermore, the later stage of the infection process - the expulsion of cargo through a 100 nm tube - remains poorly understood from a physical hydrodynamics perspective, especially when we consider the low-Reynolds number nature of the flows inside the PT.

Fluids behave in fundamentally different ways as the length scale in a physical phenomenon changes. Thus, it is critical to examine the role of physical hydrodynamics at the length scales of a single microsporidian PT by looking at the relevant dimensionless numbers. Reynolds number quantifies the relative importance of inertia and viscous force in fluid flow. When the Reynolds number is low, it means the effect of inertia is negligible compared to the viscous effect, and it is impossible to drive fluid motion without boundary movements or an external driving force (*Kundu et al., 2015*). From the geometry of the spore and the kinematics of the firing process, we can estimate the upper bound of the Reynolds number (Re) of the germination process as $\text{Re} = \frac{\rho U L}{\mu} = 3 \times 10^{-5} - 0.018$. Here $\rho$, $U$, $L$, and $\mu$ stand for the mass density of fluid (1000 kg/m³), characteristic velocity (300 μm/s), characteristic length scale, and viscosity (0.001 Pa-sec), respectively. The lower bound and upper bound of Reynolds number are computed by using PT diameter (100 nm) and full PT length (60 μm, the largest length scale) as the characteristic length scale, respectively. Since even the upper bound estimate of Reynolds number falls within the low Reynolds number regime (Reynolds number smaller than $\mathcal{O}(1)$), we expect the PT firing process will always be in the low Reynolds number regime. At this Reynolds number regime, the fluid flow will stop within $10^{-9}$ to $10^{-4}$ s once the boundary movement stops (in this case when the PT is completely ejected) and the driving force disappears (*Purcell, 1977*). This dramatic difference from inertia-dominated flows highlights the necessity to take a quantitative approach, accounting for both the low-Reynolds-number physics and experimental evidence when studying the PT firing mechanism.

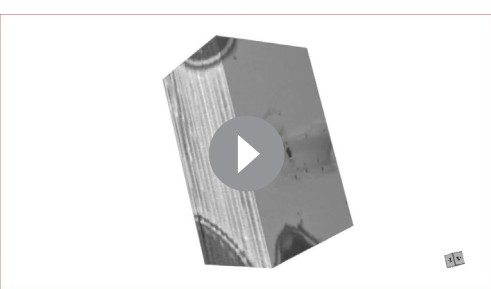

**Video 1.** 3D reconstruction of ungerminated *A. algerae* spore from SBF-SEM data. Representative 3D reconstruction of an ungerminated *A. algerae* spore. At the beginning of the video, slices through the spore are shown. Each color represents an individual organelle: exospore (orange), endospore (yellow), PT (blue), posterior vacuole (red), and anchoring disc (green). https://elifesciences.org/articles/86638/figures#video1

In this study, we perform a systematic analysis on the energy cost of the PT ejection process in microsporidia. We take a data-driven approach to generate models for the physical basis of the PT extrusion process and cargo transport through the PT. We use Serial Block-Face Scanning Electron Microscopy (SBF-SEM) to obtain three-dimensional (3D) reconstructions of spores in different stages of germination, from which we can observe snapshots of the PT ejection process. By analyzing energy dissipation in various parts of the process, we propose a model for how infectious cargo can be ejected while the PT is fully extruded - elucidating the physical principles of how infectious cargo can flow through the narrow PT (*Keeling and Fast, 2002*) in a low Reynolds number context. Our approach lays the foundation for a quantitative biophysical analysis of the microsporidian infection process.

## Results
### 3D reconstructions of spores in different stages of germination

In order to better understand the physical process of PT extrusion, and changes in PT conformation during the extrusion process, we used SBF-SEM to capture 3D snapshots of spores in different stages of PT extrusion. To this end, *A. algerae* spores were purified, activated to trigger PT extrusion by adding germination buffer, fixed, and imaged using SBF-SEM. From the SBF-SEM data, we obtained 3D reconstructions for spores in different configurations, which may represent different stages of germination. We randomly selected spores and categorized them into three states: (1) ungerminated, in which the entire PT is coiled inside the spore; (2) incompletely germinated, in which the PT is partially extruded from the spore; and (3) germinated, in which the PT is extruded, and no PT remains within the spore. Using segmentation analysis to trace the PT and all other identifiable organelles, we reconstructed 3D models of 46 spores across the three different states. These 3D reconstructions reveal the geometry of the PT and its spatial relationship to other organelles such as the posterior vacuole, anchoring disc, spore wall, and nuclei (*Figure 1B–D*). In the ungerminated spore, the anterior end of the PT is straight and attached to the anchoring disc, while the rest of the tube is coiled within

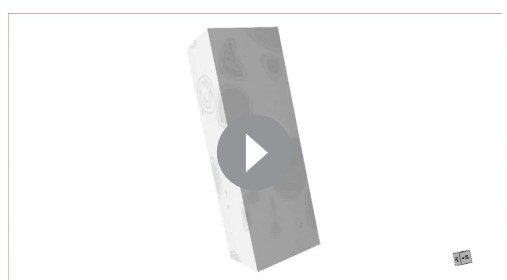

**Video 2.** 3D reconstruction of an incompletely germinated *A. algerae* spore from SBF-SEM. Representative 3D reconstruction of an incompletely germinated *A. algerae* spore. At the beginning of the video, slices through the spore are shown. Each color represents an individual organelle: exospore (orange), endospore (yellow), PT (blue), posterior vacuole (red), nuclei (pink) and anchoring disc (green). https://elifesciences.org/articles/86638/figures#video2

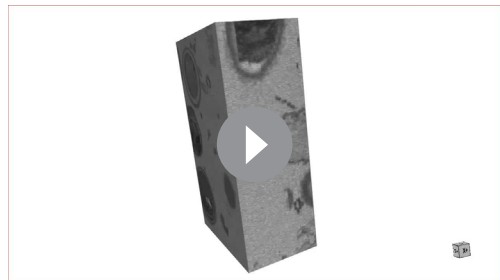

**Video 3.** 3D reconstruction of a germinated *A. algerae* spore from SBF-SEM. Representative 3D reconstruction of a germinated *A. algerae* spore. At the beginning of the video, slices through the spore are shown. Each color represents an individual organelle: exospore (orange), endospore (yellow), PT (blue), and posterior vacuole (red). Note buckling of the spore body after cargo has been expelled. https://elifesciences.org/articles/86638/figures#video3

the spore, as previously observed (*Jaroenlak et al., 2020*, *Figure 1B*, *Video 1*). The posterior vacuole sits at the posterior end and is surrounded by the coiled PT. 3D reconstructions of incompletely germinated *A. algerae* spores show the PT passing through the anchoring disc, and a rearrangement of other organelles in the spore (*Figure 1C*, *Video 2*). Germinated spores are largely empty, and contain one major membrane-bound compartment, consistent with the posterior vacuole. In addition, most germinated *A. algerae* spores are buckled, resulting in a bean-like shape (*Figure 1D*, *Video 3*).

## Systematic evaluation of possible topological configurations of a spore

While SBF-SEM data provide insights into spore organization at the organelle level, the resolution is not sufficient to ascertain the exact topological connectivity between these individual organelles. For example, even though the spatial proximity between the PT and posterior vacuole is clear, whether the end of the PT permits fluid flow between these compartments remains uncertain. To build a physical framework for the PT ejection process, it is critical to know the topological connectivity between different organelles, as the connections between organelles will determine the boundaries in the system, affecting the fluid flow and energy dissipation. Thus, we systematically evaluate the possible topological connections between organelles relevant to energetics calculations (*Figure 2*, *Supplementary file 1*). We consider six key questions to cover all hypotheses, and develop a nomenclature to describe them - (1) whether the entire tube shoots out as a slender body like a jack-in-the-box ('J'), or in a tube eversion mode ('E') in which the PT turns inside out and thus only the tip region is moving during the ejection process. Note that we use the term 'jack-in-the-box' only to describe the movement of PT, not the PT with its tip connected to cargo as in original references (*Dissanaike and Canning, 1957*). (2) whether the original PT content is open to the external environment post anchoring disc disruption or not ('OE' vs 'NOE'), (3) whether the posterior vacuole expands during the ejection process ('ExP' vs none), (4 & 5) whether the original PT content is connected to the sporoplasm ('PTS'), posterior vacuole ('PTPV'), or neither ('PTN'), and (6) whether the original PT space permits fluid flow ('none'), or is closed and cannot permit fluid flow ('PTC'). Here, we define the original PT contents as anything that is filled inside the PT before any infectious cargo enters the PT space. As the germination process progresses, the PT space does not necessarily maintain spatial proximity with the originally connected organelle. Also, when we describe a space to be connected or open to another space, it simply means that there can be fluid flow from one space to the other and cause energy dissipation (see Glossary).

Based on this nomenclature, 6 binary choices exist, leading to a total of 64 ($2^6$) possible topological configurations. We next evaluate each combination to see if it is compatible with experimental PT firing outcomes or if it is incompatible topologically. For example, the hypothesis 'J-NOE-PTN' is incompatible with experimental PT firing outcomes, as it creates an isolated PT space that would hinder the passage of infectious cargo. Another example, 'J-OE-PTS-PTC' is topologically incompatible by itself, as it is contradictory to have a PT space that is open to the external environment but is closed and cannot permit fluid flow. We apply the same compatibility criteria to these different combinations and arrive at 10 possible configurations, which also include the historically proposed mechanisms (*Ohshima, 1927*; *Weiser, 1947*; *Dissanaike and Canning, 1957*; *Keeling and Fast, 2002*; *Findley et al., 2005*; *Lom and Vavra, 1963*) as listed in *Supplementary files 1 and 2*. Based on previous imaging of the vacuole during germination (*Troemel and Becnel, 2015*) and consistent with results from volumetric reconstructions of the SBF-SEM data, we observe that the posterior vacuole volume expands during the germination process (*Figure 2—figure supplement 1*). This rules out the five configurations that assume a posterior vacuole that does not expand, leaving only five viable hypotheses (*Figure 2*). For better readability, in the following sections we refer to these five hypotheses as Model 1 through Model 5, with their abbreviation and full meaning described in the figure.

## Developing a mathematical model for PT energetics

To uncover the dynamics of the PT ejection process, it is valuable to understand energy dissipation mechanisms in organelles associated with the PT. Cargo ejection involves the spore's cellular contents traveling through a 100-nanometer-wide tube at high velocities. To better understand this, we explore hydrodynamics energy dissipation in this ultrafast process for the five viable hypotheses proposed above. Other possible sources of energy dissipation, such as the plastic deformation of the PT, will be addressed in the Discussion section. In the following, we provide a high-level summary of our

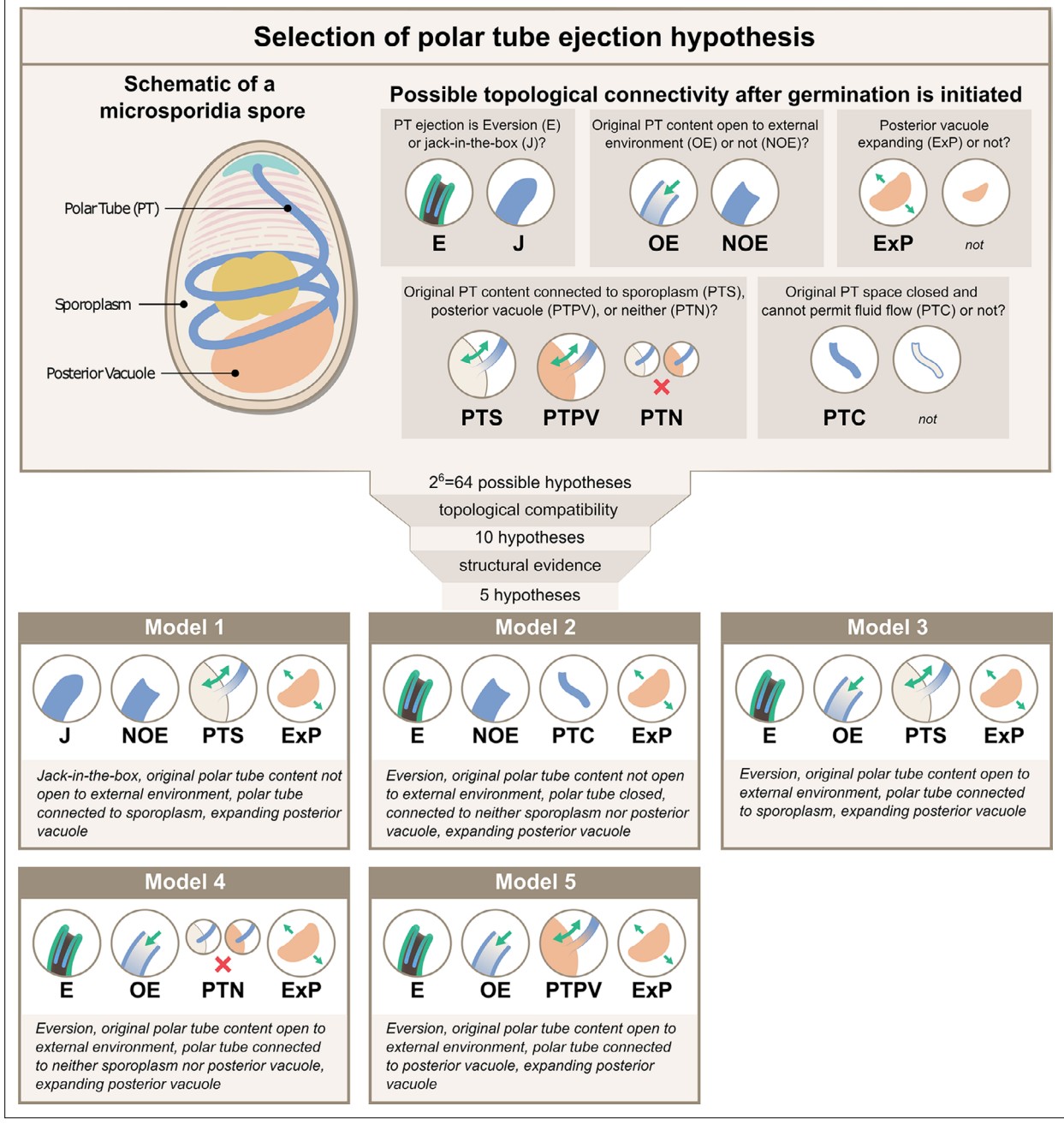

**Figure 2.** Possible hypotheses for the topological connectivity and morphology of spore organelles. The selection process of the hypotheses for the energetics calculation is shown. We considered 6 critical topological questions regarding the connections between different spaces in the spore that is relevant to the energetics calculation and developed a standard nomenclature to describe the hypotheses. The combinatorics of the 6 questions gave us 64 hypotheses. By evaluating the topological compatibility of these combinations, we are left with 10 hypotheses, and we further narrow this down to 5 hypotheses based on the fact that the posterior vacuole expands during the germination process (see *Figure 2—figure supplement 1*). The list of all the hypotheses is summarized in *Supplementary file 1*, and a detailed calculation of each hypothesis is described in *Figure 3—figure supplements 1–3*.

The online version of this article includes the following figure supplement(s) for figure 2:

**Figure supplement 1.** Volume of posterior vacuole in ungerminated and germinated spores.

calculation, with detailed derivations provided in Appendix Section A.9. Here, we do not account for the twofold length changes of PT before and after germination. The model, nonetheless, can be easily modified to account for this (see Appendix Section A.9.4). We have reported the results in *Supplementary file 7*, and the overall ranking among the proposed five hypotheses does not change.

In our calculations, we start with three sources of energy dissipation – (1) external drag (energy dissipation between a moving PT and the surroundings), (2) lubrication (energy dissipation associated with fluid flow in a thin gap), and (3) cytoplasmic flow (energy dissipation associated with fluid flow in a tube or pipe) (*Figure 3—figure supplements 1–3*). In the external drag term ($\mathcal{D}_{\dot{W}}$), we calculate the drag along the entire PT for Model 1 because in the jack-in-the-box mode of ejection, the entire tube is assumed to shoot out as a slender body. For the other four hypotheses which assume a tube eversion mechanism, only the drag at the moving tip is considered since that is the only region that is moving against the surroundings. As the drag force is linearly proportional to velocity ($v$), length scale ($l$), and surrounding viscosity ($\mu_{\text{surr}}$) in low Reynolds number regimes, and the power is the product of force and velocity, the external drag term is proportional to the square of the velocity ($\mathcal{D}_{\dot{W}} \propto \mu_{\text{surr}}v^2l$).

We next consider the energy dissipation via lubrication ($\mathcal{L}_{\dot{W}}$). First, we account for lubrication in the PT pre-eversion. Cross-sections from previous TEM studies have shown that the PT is likely composed of concentric layers (*Xu and Weiss, 2005*). Here, we account for lubrication between the two outer-most layers. Second, we include the lubrication between the uneverted part of the tube (blue) and the everted tube (green) for Model 2 - Model 5 (the four hypotheses with tube eversion mode). Finally, for Model 5, we also consider the lubrication between cargo and everted PT. We consider this because this hypothesis requires both original PT content and posterior vacuole to be open to the external environment but not to the sporoplasm, and this topology requires the cargo to be separated from the PT by a fluid gap that is connected to the fluid in external environment. The dissipation power is in the form of $\mathcal{L}_{\dot{W}} = \pi\mu_{\text{cyto}}\left(\frac{v}{h+2\delta}\right)^2 L(2Rh + h^2)$, proportional to the square of shear rate ($\dot{\gamma}^2 \propto (v/(h+2\delta))^2$) times the volume of the gap zone ($\pi L(2Rh + h^2)$). $L$ is the length of the lubrication overlapping; $R$ is the radius of the PT; $h$ is the thickness of the gap; $\delta$ is the slip length of the boundary.

In the cytoplasmic flow term ($\mathcal{C}_{\dot{W}}$), the dissipation power also scales to the square of shear rate times the volume of dissipative fluid. The shear rate is approximately the relative velocity divided by the radius (with or without slip length $\delta$) ($\dot{\gamma} \propto v/(R+\delta)$), while the volume is proportional to length times the square of radius. After multiplication, the radius terms roughly cancel each other out in power, and the final dissipative power is proportional to the square of velocity, length scale and viscosity ($\mathcal{C}_{\dot{W}} \propto \mu_{\text{cyto}}Lv^2R^0$). The detailed calculation of each term and relevant length scales are included in the lower right corner of *Figure 3—figure supplement 1*. For each observed spore germination event, we can compute the peak power requirement, peak pressure difference requirement, and total energy requirement of the PT firing process for each hypothesis, according to the equations we formulated in *Figure 3—figure supplements 1 and 2*. Note that in this work, we did not calculate the detailed pressure field around each structure. We estimated the required pressure differences between the spore and the PT tip to overcome the drag force and drive fluid flow in various spaces. Also, the same pressure differences can be achieved by either positive pressure (the spore has a higher pressure than the ambient, pushing the fluid into PT) or negative pressure (the PT tip has a lower pressure than the ambient, sucking the fluid from the spore). Hydrodynamic dissipation analysis alone cannot tell the differences between positive or negative pressure.

Since some of the energy is dissipated by internal and external fluids surrounding the spore - as listed in dissipation equations in *Figure 3—figure supplements 1 and 2* - computation of energy, power and pressure are naturally dependent both on surrounding viscosity and cytoplasmic viscosity. Note that we use the term 'cytoplasmic viscosity' as an effective viscosity for the energy dissipation within the spore, and we are not referring to the viscosity of any particular space within the spore. However, there is no reported measurement regarding the cytoplasmic viscosity of any microsporidian species so far, and previously reported values of cytoplasmic viscosity in other cell types fall into a very wide range (*Verkman, 2002*; *Luby-Phelps, 1999*; *Ridgway et al., 2008*; *Swaminathan et al., 1997*; *Brown, 1940*; *Kamitsubo et al., 1989*; *Kalwarczyk et al., 2012*; *Wang et al., 2019*). We therefore first computed the result assuming the cytoplasmic viscosity to be 0.05 Pa-s (*Brown, 1940*), a middle ground value based on the previously reported range in other cell types, and we later re-calculated our predictions using different cytoplasmic viscosity values covering the entire reported range, to assess how much our results vary depending on the degree of uncertainty in the value of cytoplasmic

viscosity. We measured the viscosity of the germination buffer and modified formulations using a commercial rheometer (*Figure 3D*, see Materials and methods section for details).

Another parameter that appears in the model is the boundary slip ($\delta$), which describes the behavior of the fluid velocity profile near a solid wall. When the boundary slip is zero (also known as no-slip boundary condition), the fluid has zero velocity relative to the boundary. As previous structural studies *Takvorian et al., 2020* have shown, an extremely thin gap (15–20 nm) may exist between the PT wall and contents inside the tube. At such small length scales, it is possible that the system can approach the continuum limits in hydrodynamic theory, which means the common assumption of no-slip boundary condition on the surface might not be valid. We therefore look at Knudsen number (defined as the ratio of molecular mean free path to the associated length scale in the problem) to check if we need to account for this effect. As the mean free path of liquid water molecules is roughly 0.25 nm (*Pennycuick, 1992*), and the thin gap between cargo and PT wall is about 20 nm, the Knudsen number is about 0.01, which is on the border between the continuum flow regime and the slip flow regime (*Pivkin et al., 2005*). The intermediate Knudsen number requires us to also perform simultaneous sensitivity testing on the slip length of the boundary. In the following section, we thus first computed the result assuming a zero slip length, and we later re-calculated the results with non-zero slip lengths.

## Theory-guided experiments differentiate between leading hypotheses

As enumerated in *Figure 3—figure supplements 1–3*, the five hypotheses listed have different contributions from the drag, lubrication and cytoplasmic flow terms, and they predict different energy requirements from the same observed firing kinematics. As each term scales differently with surrounding viscosity, changing surrounding viscosity also changes the relative magnitude of each term. Assuming that the microsporidian spores do not have spare energy generation mechanisms, we expect that as we change the surrounding viscosity, the PT firing kinematics should adjust in a way that keeps the total energy requirement the same, and thereby allow us to differentiate between the five leading hypotheses under consideration. For example, we would expect that in a jack-in-the-box ejection mechanism, increasing the surrounding viscosity should slow down the PT velocity, as the entire PT would experience changes in drag. On the other hand, a PT eversion mechanism would show less (if any) change in PT ejection velocity, since only the tip region would experience changes in drag. To differentiate between these mechanisms, we used high-speed light microscopy to observe the kinematics of *A. algerae* spore germination in buffers with varying viscosity (*Videos 4 and 5*). We used a range of methylcellulose concentrations (up to 4%) to vary the external viscosity by multiple orders of magnitude in these experiments. Changing surrounding viscosity should not change the amount of energy stored inside a spore. This is because the energy source is internal to the spore, and under our experimental conditions, the osmotic pressure change in spores due to the addition of methylcellulose is estimated to be less than 0.2% (see Materials and methods section for more detail). If a hypothesis predicts variable energy requirements based on the observed kinematics in response to changing the surrounding viscosity (statistical testing will give a *p*-value less than 0.05), that would indicate the hypothesis is not consistent with the experimental observations (*Figure 3A*). On the other hand, for a hypothesis that is consistent with experimental observations, the predicted energy requirement will not depend on the surrounding fluid viscosity (statistical testing will give a $p$-value greater than 0.05). The peak pressure difference requirement and peak power requirement are also calculated to quantitatively understand the process, but their statistics are not used for the ranking of hypotheses.

*Figure 3B* shows the observed PT length of *A. algerae* spores as a function of time in six different concentrations of methylcellulose. We found that changing the methylcellulose concentration in germination buffer up to 4%, which corresponds to an increase in viscosity of $10^3$, does not change the germination rate (*p*-value of logistic regression=0.085, see *Supplementary file 3*), maximum length of the PT (p=0.743, Kruskal–Wallis test, see *Figure 3—figure supplement 6*), or the peak velocity of PT ejection (p=0.848, Kruskal–Wallis test, see *Figure 3C*). The observation that there is no change in velocity of PT firing regardless of external viscosity provides qualitative support to the four hypotheses utilizing an eversion mechanism over the jack-in-the-box ejection mechanism. The full original data can be found in *Figure 3—figure supplement 5*.

For each observed spore germination event, we next computed the peak power requirement, peak pressure difference requirement, and total energy requirement of the germination process for each

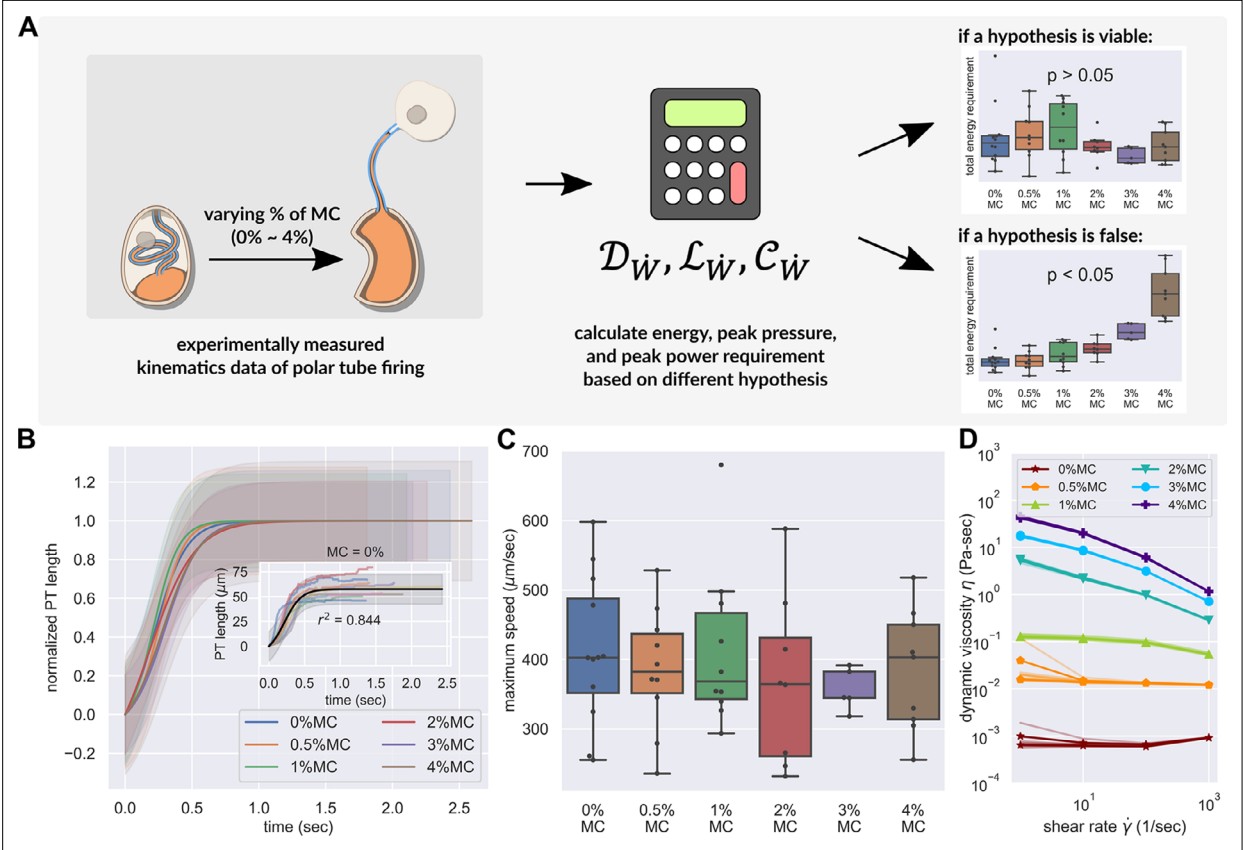

**Figure 3.** PT firing kinematics in the presence of varying external viscosity. (**A**) Schematic outlining the protocol for hypothesis testing. We experimentally measured the PT firing kinematics of *A. algerae* spores in buffers with varying viscosity, by varying the methylcellulose (MC) concentrations up to 4% (***Videos 4 and 5***). We next calculated the required total energy, peak pressure and peak power for each experimentally measured data according to our physical framework proposed in ***Figure 3—figure supplements 1–3***, and we see if the required energy changes with respect to changes in surrounding viscosity. We assume that changing surrounding viscosity should not change the energy sources of the spores. Thus if the calculated energy requirement changes significantly with respect to changes in surrounding viscosity (p<0.05), the hypothesis is inconsistent with experimental observations. (**B**) Experimental measurement of PT ejection kinematics of *A. algerae* spores in different concentrations of methylcellulose. The kinematics was fit to a sigmoid function $y = L(\frac{1}{1 + e^{-k(x-x_0)}} - \frac{1}{1 + e^{kx_0}})$ and then normalized by $L$. The additional term in the sigmoid function is to ensure the curve passes the origin. (0%: n=12; 0.5%: n=10; 1%: n=10; 2%: n=8; 3%: n=5; 4%: n=9) The inset shows the original data in MC0%. The changes in MC concentration does not cause obvious changes in overall kinematics of PT firing. The complete set of original data can be found in ***Figure 3—figure supplement 5***. (**C**) The dependence of maximum PT ejection velocity on MC concentration in germination buffer. Increasing MC concentration up to 4% does not change the maximum PT ejection velocity. (p=0.848, Kruskal–Wallis test) (**D**) Viscosity measurements of germination buffer with various concentrations of methylcellulose, corresponding to the concentrations used in PT extrusion experiments. As the PT ejection process is a high shear rate phenomenon (~3000 1 /s), we used the measurement at shear rate $\dot{\gamma} = 1000$ s$^{-1}$. The maximum tested shear rate was 1000 s$^{-1}$ as that reaches the operation limit of the shear rheometer. (n=5 for 0%, 0.5%, 1%. n=3 for 2%, 3%, 4%.).

The online version of this article includes the following figure supplement(s) for figure 3:

**Figure supplement 1.** Calculations for energy dissipation of the PT firing process.

**Figure supplement 2.** Calculations for the required pressure differences of the polar tube (PT) firing process.

**Figure supplement 3.** Flow fields used for energy dissipation calculation in ***Figure 3—figure supplement 1*** and ***Figure 3—figure supplement 2***.

**Figure supplement 4.** Evaluation of the experimental challenges of shear rheology in the measurement of buffer viscosity.

**Figure supplement 5.** Experimental measurement of PT ejection kinematics of *A.algerae* spores in different concentrations of methylcellulose.

**Figure supplement 6.** Dependence of maximum PT length on the methylcellulose concentration in germination buffer.

hypothesis (***Figure 4***). Assuming a cytoplasmic viscosity of 0.05 Pa-s and a no-slip boundary condition, we can see that Model 1 (***Figure 4A***) and Model 3 (***Figure 4C***) contradict our experimentally observed PT firing kinematics. Model 1 predicts a significant increase in total energy requirement, which cannot be explained by the observed kinematics. On the other hand, Model 3 predicts a total

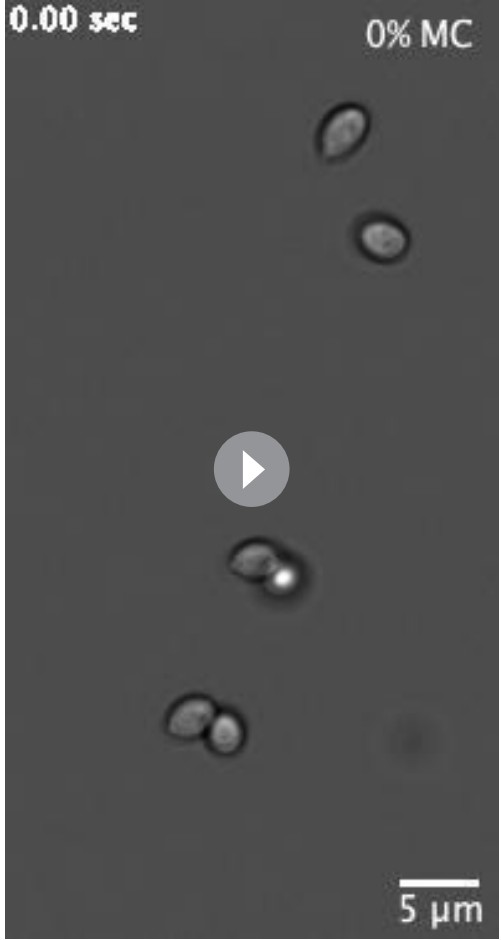

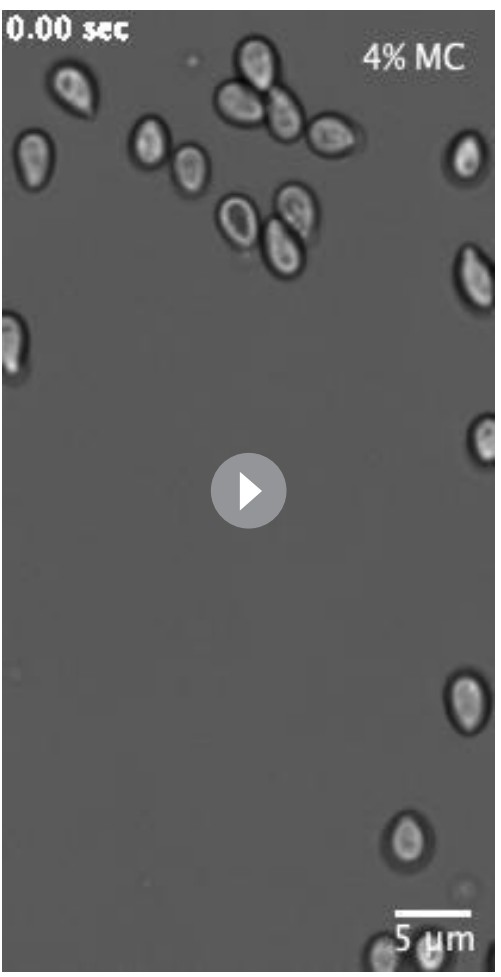

**Video 4.** Live-cell imaging of *A. algerae* PT germination in 0%MC.
https://elifesciences.org/articles/86638/figures#video4

**Video 5.** Live-cell imaging of *A. algerae* PT germination in 4%MC.
https://elifesciences.org/articles/86638/figures#video5

energy requirement that varies substantially and is inconsistent with the experimentally observed data. It is worth noting that for the remaining three viable hypotheses (Model 2, Model 4, and Model 5), the total energy requirement is roughly $10^{-11}$ J, the peak pressure difference requirement is roughly 60–300 atm, and the peak power requirement is roughly $10^{-10}$ W, all in a very similar range. As a comparison, an *E. coli* swimming in water for 60 μm at a speed of 25 μm/s would only cost an energy of $2.8 \times 10^{-17}$ J (calculated from Stokes drag, assuming a characteristic length of 1 μm), a much smaller number. The huge difference in energy requirement is consistent with the physical intuition that the high speed and high resistance experienced by fluid flow during germination makes the ejection process energetically costly. It is interesting that our calculated pressure is comparable to other biological phenomena where pressure is relevant. For example, the pressure difference requirement is comparable or greater than that required for DNA packaging in phages (roughly 60 atm *Smith et al., 2001*).

As mentioned earlier, the above calculation requires the exact knowledge on cytoplasmic viscosity, which has never been characterized for microsporidian species. We therefore repeat the same set of calculations with varying cytoplasmic viscosity ranging from 0.001 Pa-s, 0.05 Pa-s, 0.8 Pa-s, and 10 Pa-s (informed by a range of viscosity measurements across eukaryotic species). As we previously described, changing surrounding viscosity should have no effect on how much energy a spore can generate, and thus a statistical test should report a $p$-value greater than 0.05 if the physical mechanism is consistent with experimental observations. As shown in *Supplementary file 4*, all the calculations that differ significantly from expectation come from Model 1 and Model 3, indicating that these models are the least likely mechanisms of PT firing. However, if the cytoplasmic viscosity is too high,

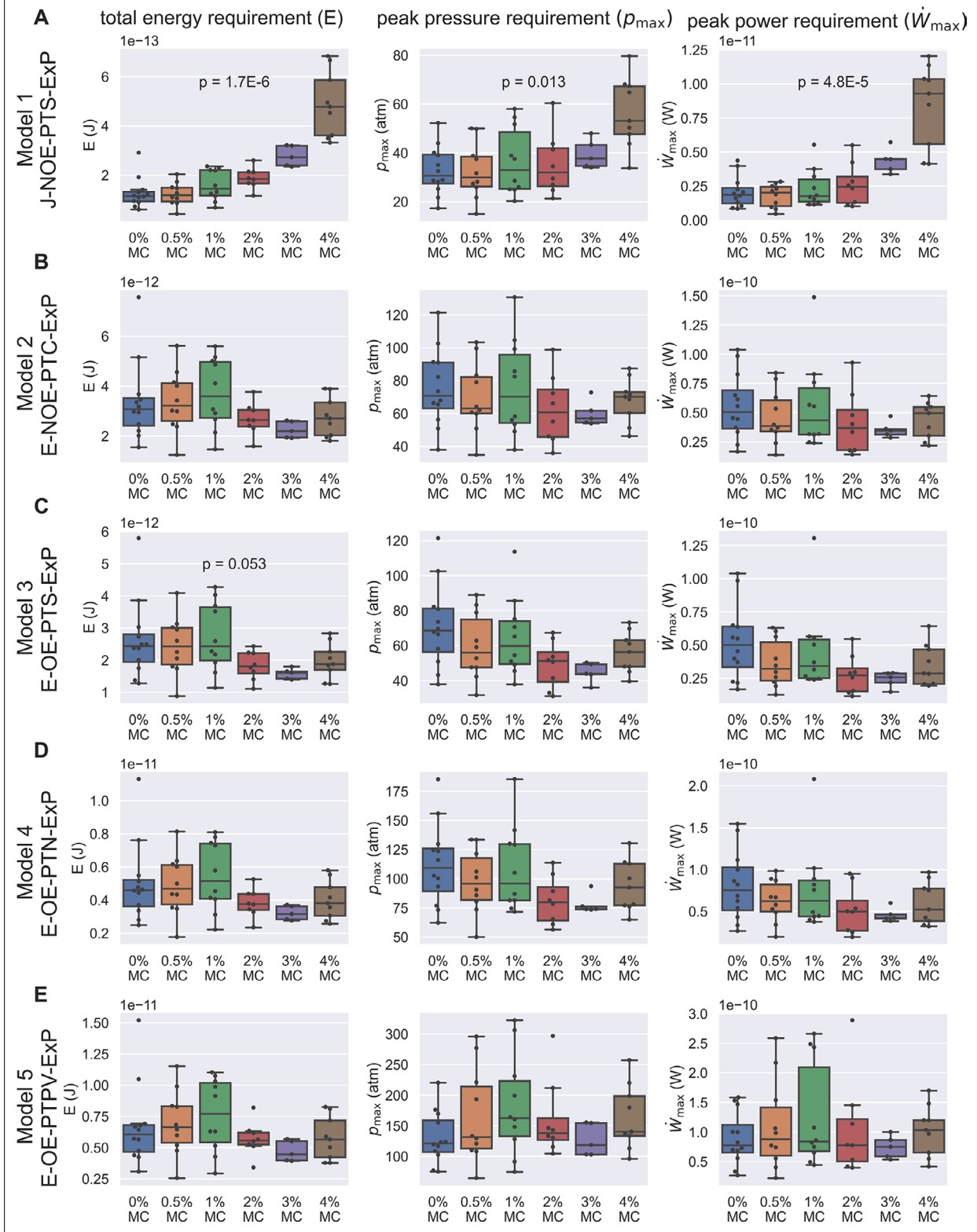

**Figure 4.** Energetic analysis to identify hypotheses that are consistent with experimental results of PT extrusion kinematics in varying external viscosities. Each row (**A–E**) shows calculations based on the five different hypotheses, and the three columns show the calculation for total energy requirement (left column), peak pressure difference requirement (middle column), and peak power requirement (right column) for each PT firing event shown in *Figure 3—figure supplement 5*. Kruskal–Wallis test was used, and only the *p*-values which are significant or near-significant are shown. Only the *p*-values calculated for total energy requirement were used for ranking. The *p*-values for peak pressure difference requirement and peak power

*Figure 4 continued on next page*

*Figure 4 continued*

requirement are just for reference. The data shown here is calculated assuming a cytoplasmic viscosity of 0.05 Pa-s, and a zero boundary slip. The effect of ambiguity in cytoplasmic viscosity and slip length of the boundaries are discussed in *Supplementary files 4 and 5*. Under these assumptions, Model 1 and Model 3 are the two hypotheses that are least likely to be true. Also note that for the other three hypotheses (Model 2, Model 4, and Model 5), the total energy requirement is roughly $10^{-11}$ J, the peak pressure difference requirement is roughly 60–300 atm, and the peak power requirement is roughly $10^{-10}$ W.

The online version of this article includes the following figure supplement(s) for figure 4:

**Figure supplement 1.** Energy breakdown of different hypotheses.

most of the energy requirement comes from the energy dissipation within the spore and PTs. In this case, changing the surrounding viscosity has little effect regardless of the mechanism, and therefore cannot help differentiate the hypotheses. Thus the effectiveness of our experimental design in differentiating the five hypotheses changes as a function of cytoplasmic viscosity.

Next we consider the role of boundary slip. As discussed earlier, the intermediate Knudsen number requires us to also perform simultaneous sensitivity testing on slip length of the boundary. Therefore, we repeated the calculation in *Supplementary file 4* (which corresponds to a slip length=0 nm, or no-slip boundary condition) with slip length=15 nm or 60 nm. We cap our calculation at slip length of 60 nm as that is three times larger than the dimension of the gap, and further increasing the slip length would have little effect. As shown in *Supplementary file 5*, Model 1 and Model 3 remain the two most likely rejected hypotheses as we change the slip length of the boundary and the cytoplasmic viscosity. If the cytoplasmic viscosity is 0.001 Pa-s and the slip length equals 15 nm, Model 2 is also rejected. Note that in the limit of large slip length and low cytoplasmic viscosity, all five hypotheses will be rejected, because in this case there is essentially no dissipation from the fluid inside the spore. All the energy dissipation will then scale unfavorably to changes in surrounding viscosity, and thus cannot explain the observed kinematics in our experiments. This methodology does not differentiate between Model 4 and Model 5 - and they remain preferred over the other three hypotheses.

Our model allows us to differentiate between different hypotheses based on kinematic observations, a readily accessible experiment. Furthermore, we can also analyze the relative contributions of various dissipation terms, which would not be possible to measure experimentally. As an example, in *Figure 4—figure supplement 1A*, we show why Model 1 and Model 3 are rejected in our baseline case ($\mu_{\text{cyto}}$=0.05 Pa-s, $\delta$=0 nm). For Model 1, the external drag term scales up unfavorably with changes in surrounding viscosity, which is expected as the slender body theory predicts a drag force that roughly scales linearly with the length of the PT. For Model 3, the lubrication that is accounted for in the model is not enough to buffer out the variations in experimental observation and is therefore also rejected. Compared to Model 1 and Model 3, Models 4 and 5 do not have an external drag term that scales up unfavorably with changes in surrounding viscosity. These two hypotheses (Model 4 and Model 5) are not rejected as they account for enough terms in cytoplasmic flow and lubrication to buffer out the variations in experimental observation. In our slip boundary case with low cytoplasmic viscosity ($\mu_{\text{cyto}}$=0.001 Pa-s, $\delta$=15 nm), Model 1, Model 2, and Model 3 are all rejected (*Figure 4—figure supplement 1B*). In this scenario, the energy dissipation from fluid inside the spores is greatly reduced and the contribution from external drag becomes more prominent. Model 1 is rejected because of similar reasons as mentioned before. For Model 2 and Model 3, not enough energy dissipation terms are accounted for, which fails to buffer out the unfavorable scaling of external drag with changes in surrounding viscosity.

## Models for the driving force behind cargo expulsion

The primary function of the PT is to transport infectious cargo into the host cell. A unique two-stage process of nuclear translocation was recently observed using high-speed imaging (*Jaroenlak et al., 2020*), wherein the nuclei, ~1 μm in diameter, are grossly deformed to pass through the ~100-nm-wide PT. Instead of traveling smoothly to the end of the PT, the nucleus pauses in the middle of the tube and is then abruptly expelled from the end (*Figure 5A–B*). Previous imaging studies also demonstrate that nuclear translocation is not initiated until 50% of the PT has been ejected (*Weidner et al., 1994*; *Weidner et al., 1995*; *Jaroenlak et al., 2020*). However, since the PT firing process is a low Reynolds number event with no inertial terms, it is impossible to push any cargo or cytoplasmic content inside the PT any further once the extension of PT stops without invoking additional mechanisms or energy

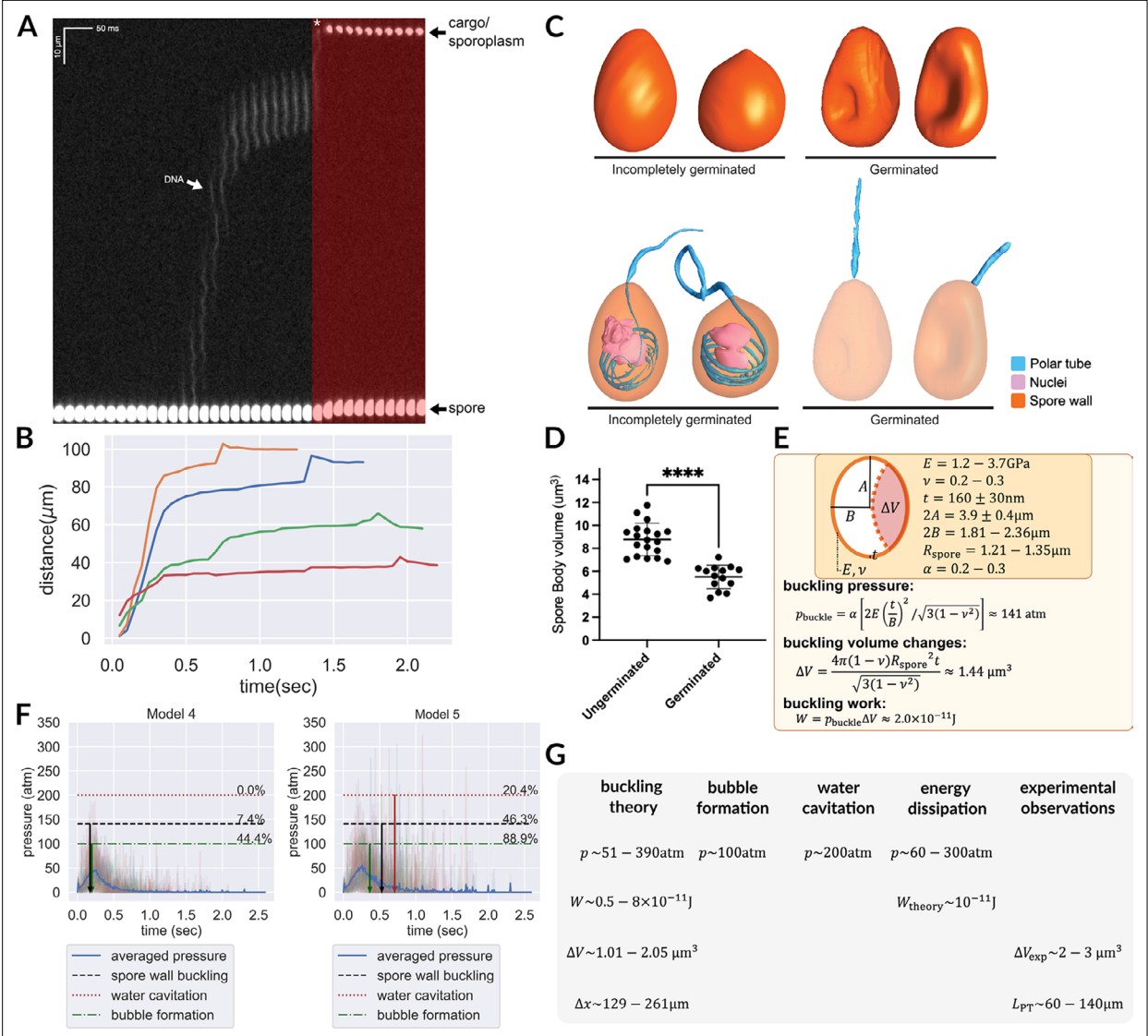

**Figure 5.** Hypotheses that can potentially explain the two-stage translocation of the cargo. (**A**) Kymograph of nuclear transport inside the PT. Nuclei were stained with NucBlue prior to germination, and imaged using fluorescence microscopy. Previously deposited data from *Jaroenlak et al., 2020* were used in this figure. A two-stage process is observed for nuclear translocation, with a long pause in the middle. The second stage of nuclear movement is overlaid with red, and the asterisk indicates the beginning of the second stage movement, in which the nuclei are expelled out of the PT. (**B**) Quantification of the nuclear position relative to spore coat over time (n=4). (**C**) 3D reconstructions of incompletely germinated and germinated spores from SBF-SEM data. 100% of spores in which the nuclei have been expelled are buckled (*Supplementary file 6*). The translocation of nuclei at the final stage can be explained by spore buckling. (**D**) Volumes of ungerminated and germinated spores calculated from SBF-SEM 3D reconstructions. Ungerminated: mean=8.78 µm³, std=1.41 µm³, n=19; Germinated: mean=5.52 µm³, std=1.03 µm³, n=14; p<0.0001. (**E**) Schematic model of an *A. algerae* spore used for calculating the spore wall buckling pressure, the relevant parameters used in the calculation and the formulae. Using the theory of elastic shell buckling (see text for detail), we showed that the pressure built up during the PT firing process is enough to buckle the spore wall, and the predicted buckling volume is enough to push cytoplasmic content in PT forward by 129–261 µm. (**F**) The predicted time series of pressure from Model 4 and Model 5 (n=54), overlaid with the critical pressure of spore wall buckling, water cavitation pressure and bubble nucleation. All three phenomena can cause volume displacement at the later stage of the germination process, and provide a driving force to push the cargo/nuclei forward. Model 5 is more compatible with experimental data than Model 4. The downward arrows indicate the mean time when the negative pressure first reaches the critical pressure. (detailed numbers mentioned in the main text.) (**G**) Theoretical predictions and experimental measurements from orthogonal approaches are compiled and are in agreement with each other. We obtained the prediction based on spore wall buckling theory and hydrodynamic energy dissipation theory, and we compiled the experimental observations from the SBF-SEM data. Symbols: $R_{spore}$: spore radius; $\Delta V$: volume changes of spore after buckling; $t$: spore wall thickness; $E$: Young's modulus of the spore wall; $\nu$: Poisson ratio of the spore wall; $W$: work; $\Delta x$: predicted fluid displacement distance; $L_{PT}$: full length of the ejected PT.

The online version of this article includes the following figure supplement(s) for figure 5:

**Figure supplement 1.** Dependence of spore buckling probability on the threshold pressure of spore wall buckling.

sources. Currently, our understanding of how the cargo can be forced into and through the PT and what driving forces are involved remains inadequate. Our data presented here provide two possible mechanisms for the final extrusion of cargo, which will be discussed in more detail in the subsections below: (1) buckling of the spore wall, which is also observed in our SBF-SEM data and (2) cavitation or bubble formation inside the spore.

Our SBF-SEM data provide an important clue: 88% of germinated *A. algerae* spores are buckled inwards (*Figure 5C*, *Supplementary file 6*). Out of 25 germinated spores, 22 have buckled walls. Of these 22 buckled spores, 21 contain no nuclei, while only 1 of the 22 has the nuclei inside. Only 3 out of 25 fully germinated spores do not have a buckled spore wall, and all 3 of these spores have the nuclei retained inside. Importantly, all spores in which the nuclei have been ejected have buckled walls, while all incompletely germinated spores, which contain nuclei in them, are not buckled (50 out of 50). These observations strongly suggest that spore wall buckling correlates with successful nuclear translocation. The inward buckling of the spore coat indicates that the pressure differences required during PT ejection come from a negative pressure, created inside the spore as PT leaves the spore. Here we hypothesize that this inward buckling displaces fluid to facilitate the second phase of nuclear translocation, expelling the nuclear material out of the spore. This hypothesis further allows the timing of this process to be controlled - where the negative pressure for the spore wall to buckle is only reached when the tube is extended near-completely.

We next estimated the energy and pressure that is required to buckle the spore shell utilizing classical buckling theory (*Zoelly, 1915*; *Hutchinson, 2016*), assuming a prolate spheroid shape for the spore. Using the reported Young's modulus ($E$) of chitin in literature (about 1.2–3.7 GPa *Yusof et al., 2004*), and assuming the Poisson ratio ($\nu$) to be 0.25 (as most solid materials have a Poisson ratio between 0.2–0.3 *Kaleli et al., 2018*), we calculate the negative pressure needed for spore buckling. A previous microscopy study shows that the exospore thickness ($t$) of *A. algerae* is roughly 160±30 nm, the length of the spore is 3.9±0.4 µm, and the volume of the spore is 8.8±1.4 µm³. From these numbers, the effective width of the spore used for calculation can be estimated as 1.81–2.36 µm, with an aspect ratio between 1.48–2.37. (We did not use the experimentally measured width of the spores since they are not precisely in prolate spheroid shape.) We can thus estimate the pressure, displaced volume, and work done by buckling as

$$p_{\text{buckle}} = \alpha \left[ 2E(\frac{t}{B})^2 / \sqrt{3(1-\nu^2)} \right] = 51 \sim 390 \text{ atm [mean 141 atm]}$$

$$\Delta V = \frac{4\pi(1-\nu)R_{\text{spore}}^2 t}{\sqrt{3(1-\nu^2)}} = 1.01 \sim 2.05 \mu\text{m}^3 \text{ [mean 1.44}\mu\text{m}^3]$$

$$W = p_{\text{buckle}}\Delta V = 5.2 \times 10^{-12} \sim 8.0 \times 10^{-11}\text{J [mean } 2.0 \times 10^{-11}\text{J]}$$

, where $B$ is the semi-minor axis of the ellipsoid, and α is an aspect-ratio-dependent prefactor associated with non-spherical shape. Based on previous studies (*Danielson, 1969*), α would be between 0.2–0.3 given the aspect ratio of the spore. In the calculation of buckling volume, we assumed a spherical shape and estimated the radius to be 1.21–1.35 µm, since there are no tabulated numbers of buckling volumes for non-spherical shapes. The geometric mean is used, as the range covers values of different orders of magnitude.

It is worth noting that the pressure and work fall within the predicted range shown in *Figure 4*, and the displaced volume is also in a reasonable range relative to the total volume of the spore. The estimated displaced volume is also consistent with the experimentally observed volume changes of spores after germination as measured by SBF-SEM (*Figure 5D*). Assuming that the PT is a 100-nm-diameter cylinder, this buckling event is enough to push forward the fluid content inside the PT by 129–261 µm. This distance is sufficient to propel the nucleus to travel through a completely ejected tube, whose length is between 60 and 140 µm (*Jaroenlak et al., 2020*).

While buckling of germinated spores is apparent in *A. algerae*, we also considered the possibility that some other species may have thicker cell walls, and may not buckle. Since our previous calculations, and the fact that *A. algerae* spores buckled inward, indicate that there is a large negative pressure during the germination process, we further explore the possibility of water cavitation or carbon dioxide bubble formation ("bubble formation" henceforth) inside the spore as an alternative mechanism. Both are phase transition events that can only occur under negative pressure at a certain threshold and can cause volume displacement from the spore into the PT. The threshold for water

cavitation is about −200 atm (*Herbert et al., 2006*; *Scognamiglio et al., 2018*) while the threshold for bubble formation is about −100 atm (*Harvey, 1975*). Since the pressure range seems plausible, we next combine our energy dissipation analysis with this pressure threshold to see if we can quantitatively predict the fraction of spores that can pass through the threshold, and the timing of these volume displacement events based on the experimentally observed kinematics.

*Figure 5F* shows the time series of pressure predicted by Model 4 and Model 5, the two most preferred hypotheses in our previous analysis. For each hypothesis, we calculate the fraction of spores that have their pressure exceeding the critical pressure for the second stage cargo translocation, either through spore wall buckling, cavitation or bubble formation. The downward arrows indicate the mean time when the negative pressure first reaches the critical pressure of different mechanisms. For Model 4, 44.4% of spores can have bubble formation, 7.4% of spores can have spore wall buckling, and none of them can have water cavitation. On the other hand, for Model 5, 88.9% of spores can have bubble formation, 46.3% can have spore wall buckling, and 20.4% can have water cavitation. The time series of pressure also allows us to predict the timing of this second-stage translocation event for different models. For Model 4, the predicted second-stage event happens at 0.17–0.2 s after initial germination (spore wall buckle: mean=0.173 s, std=0.020 s, n=4; cavitation: none; bubble formation: mean=0.198 s, std=0.082 s, n=24). For Model 5, the predicted second-stage event happens at 0.36–0.7 s after initial germination. (spore wall buckle: mean=0.530 s, std=0.335 s, n=25; cavitation: mean=0.709 s, std=0.392 s, n=11; bubble formation: mean=0.364 s, std=0.249 s, n=48). We can see that Model 5 compared to Model 4 has a much better prediction in terms of the fraction of spores that can undergo spore wall buckling. For Model 5, 88.9% of the spores can potentially form bubbles. On the other hand, as water cavitation requires a much higher negative pressure, the fraction of spores that can achieve this is much lower. Nonetheless, our analysis shows that this mechanism is still possible, though not the most likely. In the future, we can further test this hypothesis by recording the acoustic signal with a miniature hydrophone to detect the acoustic signature of water cavitation (*Scognamiglio et al., 2018*).

We note that even for Model 5, the predicted 46.3% buckling rate is much lower than the observed 88% buckling rate in germinated spores in SBF-SEM, yet we should also note that the range of predicted spore wall buckling threshold is very broad (51–390 atm, with 141 atm as the geometric mean, mostly from the uncertainty in the Young's modulus of the spore wall). If we set the threshold of buckling to be the minimum value in the predicted range (51 atm), then Model 4 would predict 98% spores to buckle while Model 5 would predict 100% spores to buckle. In *Figure 5—figure supplement 1* we show how the predicted buckling probability varies for Model 4 and Model 5 through the whole predicted range, and we can see that Model 5 consistently predicts a buckling rate that is closer to experimental observations over Model 4.

## Discussion

For more than a century, the process of microsporidia PT ejection has been qualitatively described. Yet, a comprehensive biophysical evaluation of the feasibility of the hypotheses and models proposed remains lacking. Despite the advances in imaging techniques (*Takvorian et al., 2020*; *Jaroenlak et al., 2020*), current data remain inadequate to decipher the topological connectivity of distinct organelles within a whole spore. Here we took a systematic approach using physical principles to validate different hypotheses on topological connectivity and energetics, both experimentally and theoretically.

### Physical benefits of ultrafast PT ejection during germination

Why did microsporidia evolve the PT ejection process to be an ultrafast event? The targets of the PT are usually not rapidly moving, why not achieve the same travel distance at a lower speed? Ultrafast PT ejection may be useful for the parasites in the context of the extracellular matrix in the host. One of the most common infection sites is the intestinal epithelium, which is covered by mucin and other complex viscoelastic fluids (*Grondin et al., 2020*). As the shear rate increases to 1000 sec$^{-1}$, comparable to the physiological shear rate generated by microsporidia, the shear viscosity of mucin solutions typically shear-thin by at least 2–3 orders of magnitude (*Sardelli et al., 2019*). This can bring down the viscosity of mucin polymer from 1 Pa-s to a viscosity that is close to water (*Curnutt et al., 2020*).

As mucin and other bio-polymeric fluids frequently exhibit shear and extensional thinning (*Ahmad et al., 2018*), an ultrafast movement of the PT and the high shear rate associated with the narrow tube diameter may help the organism to reduce resistance from the external environment. In this study, we also show that the eversion mechanism can further limit the external drag to the tip region, reducing the work that needs to be done for the infection process. Future work undertaking a full biophysical account of the energy dissipation, in combination with high-resolution structural data, will elucidate how the combination of ultrafast ejection and an extremely narrow tube can work together to the benefit of the organism.

## Energy dissipation from PT plastic deformation

Our experimental imaging, 3D reconstructions and theoretical analyses support the common consensus that PT ejection is indeed a tube eversion process. This is consistent with our observation that the shape pattern of the ejected tube (e.g. the helical or zigzag shape) remains static and does not alter between frames of the movie as the ejection progresses. As the eversion process involves a 180 degree turn and is typically described by large deformation theory, it raises the possibility of material yielding and plastic deformation, which can dissipate additional energy (*Yang et al., 2019*). From an evolutionary standpoint, it would be optimal for microsporidia to evolve its PT such that the tube would never experience plastic deformation to avoid hysteresis and ensure that the PT can always recover to its completely ejected configuration. Also, the ultrathin nature of the PT wall (roughly 5–30 nm *Takvorian et al., 2020*) can help reduce the stress associated with the bending of the tube, avoiding reaching the yield stress of the PT. Considering these arguments, and the fact that the material properties of the PT protein have not been well characterized, we did not consider this in our calculation of energy dissipation.

## Posterior vacuole expansion and the role of osmotic pressure

In this study we quantified that the posterior vacuole of *A. algerae* spores expand by roughly 0.35 $\mu m^3$ based on the 3D SBF-SEM data (*Figure 2—figure supplement 1*). This observation is consistent with previous real-time light microscopy of posterior vacuole expansion on *Edhazardia aedis* (*Troemel and Becnel, 2015*). One leading hypothesis in the field is that the energy source for germination comes from the expansion of the posterior vacuole due to osmotic pressure (*Lom and Vavra, 1963*; *Undeen, 1990*; *Undeen and Frixione, 1990*; *Undeen and Vander Meer, 1999*). In this paper, we made no assumptions on how the energy, pressure or power is generated, as further experiments and/or simulations are required to understand these processes. In the following paragraphs, we will discuss and quantitatively evaluate the possibility of posterior vacuole expansion as the energy source of the germination process.

Prior work has demonstrated the importance of osmotic pressure for the germination process. Studies have shown that increased osmotic pressure in the environment suppresses the germination of several microsporidian species. Ohshima showed that an osmotic pressure of 120 atm (15% saline) suppresses the germination of *Nosema bombycis* (*Ohshima, 1927*), while Lom & Vavra showed that an osmotic pressure of 60 atm (50% glucose) suppresses the germination of *Pleistophora hyphes-sobryconis* (*Lom and Vavra, 1963*). Undeen and Frixione also report that the PT emergence time can be prolonged from 1 to 2 s to 10–100 s under hyperosmotic conditions (*Undeen and Frixione, 1990*). Based on prior measurement of sugar content in *A. algerae* spores, we can also estimate the osmotic pressure inside the spores to be roughly 60 atm (see Method for calculation details). These experimental results suggest that osmotic pressure can play a role beyond just the initiation of the germination process, and might also drive PT extrusion.

Combining these experimental data, we can evaluate whether the expansion of the posterior vacuole due to osmotic pressure can provide enough energy for the entire germination process. The energy that can be provided by water influx causing 0.35 $\mu m^3$ volume expansion under the osmotic pressure difference of 60 atm is $(60 \text{atm})(0.35 \mu m^3) \sim 2.1 \times 10^{-12}$J. We can see that although the pressure is comparable to the peak pressure difference requirement (60–300 atm) calculated from our theory, the total energy provided is about 5-fold smaller than the total energy requirement (~$10^{-11}$J). This indicates that although posterior vacuole expansion can indeed provide a significant portion of energy, it may not be enough to sustain the entire germination process in *A. algerae*. It is still possible that for other species with larger magnitude of posterior vacuole expansion, osmotic pressure can

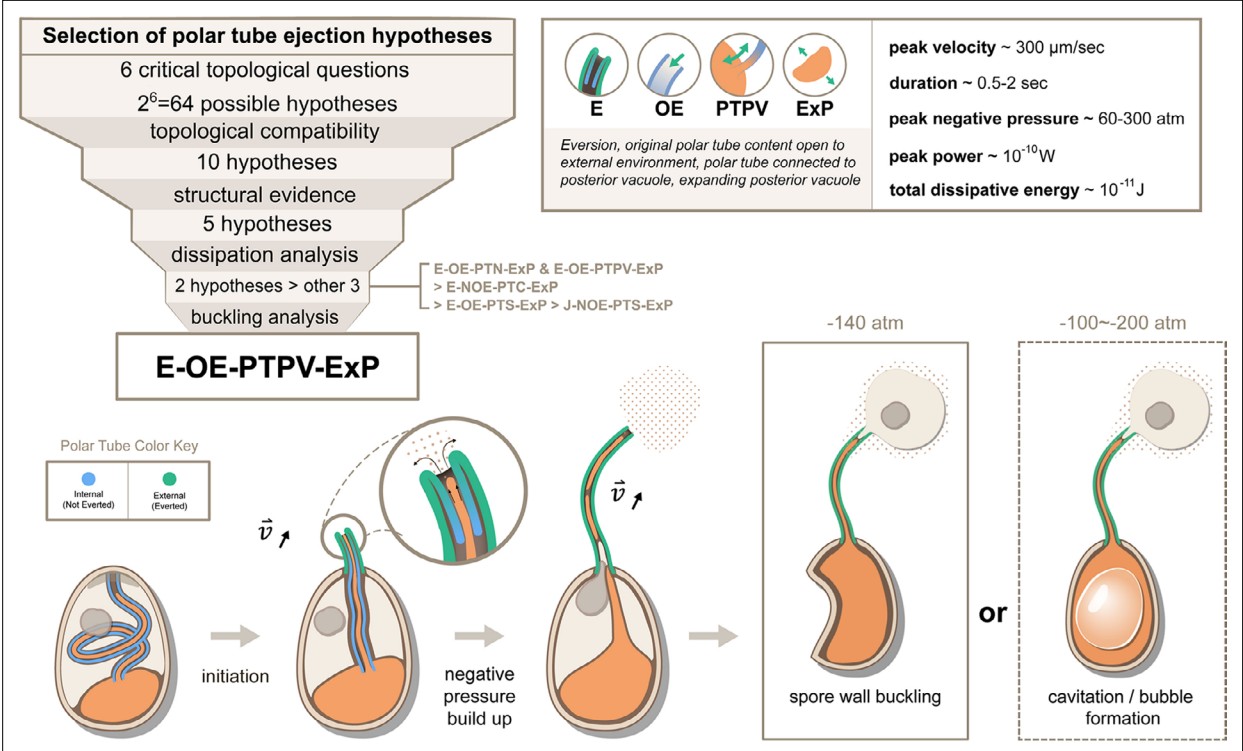

**Figure 6.** Summary and a model for the most likely hypothesis of the PT firing mechanism. We evaluated 64 possible topological connectivities, eliminated those that are incompatible with our knowledge of the process, and further explored 10 viable hypotheses. We retained the five hypotheses that assume an expanding posterior vacuole during the germination process, which are consistent with the SBF-SEM data. The hydrodynamic energy dissipation analysis allows us to rank 2 hypotheses over the other 3, and our analysis on the pressure requirement for spore wall buckling suggests Model 5 (E-OE-PTPV-ExP, 'Eversion, with PT tip open to external environment, and PT connected to posterior vacuole, with expanding posterior vacuole') is the most preferred hypothesis. The schematic shows our understanding of the process based on Model 5. After initiation of germination, the PT extrudes via an eversion-based mechanism. Vacuole contents may be connected to the original PT contents. The eversion brings the end of the PT away from the posterior vacuole, which allows the infectious cargo to later enter the PT through fluid entrainment. Tube eversion causes negative pressure to build up within the spore. Eventually this negative pressure either initiates buckling of the spore wall or causes bubble formation in the spore to push the nucleus outward. Key numbers related to the process and the predictions from E-OE-PTPV-ExP hypothesis are summarized in the text box.

play a more important role in the germination process, yet additional studies are needed to identify and quantitatively evaluate other energy sources.

## Predictions and proposed future experiments

In this study, we utilize a general framework to create the five most viable hypotheses, informed by our structural studies of the spore. Here we emphasize that our biophysical study can only provide a ranking among these five hypotheses rather than rejecting any of them explicitly. This is primarily due to lack of measurements for cytoplasmic viscosity and boundary slip length in current experiments. To deal with this ambiguity, we repeat the calculation on a wide range of possible cytoplasmic viscosity and boundary slip length to see how much our conclusion may change. Our work provides a systematic approach that can be readily adaptable as more experimental evidence comes to the table, and the general physical phenomena highlighted here would not change. Combining all evidence, our study suggests that Model 5, E-OE-PTPV-ExP ('Eversion, with original PT content open to external environment, and PT connected to posterior vacuole, with expanding posterior vacuole'), is the most preferred hypothesis (*Figure 6*). This is also consistent with the hypothesis proposed by *Lom and Vavra, 1963*. The model provides several predictions that can be readily tested by experiments. First, our model predicts that the content of the posterior vacuole should be detectable in the surroundings near the ejected tube after the germination process. This is because the original PT content (which is connected to the posterior vacuole) needs to be expelled into the surroundings before the infectious cargo can enter the PT. Second, our model predicts the relative time sequence of PT tip extension,

cargo translocation and spore wall buckling. According to our model, we should see that (1) the cargo would not enter the PT until at least half of the tube is ejected, (2) the spores only buckle during the later stage of the germination, and (3) the sudden translocation of nuclei/cargo coincides with or is slightly later than the buckling of the spore. Exploration of this hypothesis would likely require designing a custom-built microscope to simultaneously observe the kinematics of germination events at low magnification (with sporoplasm and nucleus fluorescently tagged) while having a close-up view on spore shape, to help visualize the relative kinematics. Third, the spillage of posterior vacuole content during the PT ejection event would also predict a different flow field near the tip compared to the movement of a solid boundary. Future experiments using particle image velocimetry (PIV) near the ejection tip to identify the presence of extruding fluid from the PT content will be informative. Fourth, our theory also predicts that some spores can have water cavitation inside the spore due to the large negative pressure. Using miniature hydrophone recording may capture the characteristic acoustic signal of this process if it happens. Finally, according to Model 5, the membrane connection between PT and posterior vacuole must be broken for the infectious cargo to enter the PT. There are no current data that support membrane fission in this process, but membrane fission mediated by shearing can occur on extremely fast timescales (*Morlot and Roux, 2013*). In theory, the membrane content in PT can potentially be severed into multiple parts by Plateau-Rayleigh instability, an interfacial-tension-driven fluid thread breakup mechanism. Future work will be necessary to assess whether this occurs in microsporidia, and may play a role during PT germination.

## Conclusions

In conclusion, we propose a comprehensive theoretical framework of the energy dissipation in the ultrafast PT ejection process of microsporidia, with five different hypotheses classified according to the key topological connectivity between spaces. We estimated that for the PT discharge of *A. algerae* spores, the total energy requirement is roughly $10^{-11}$ J , the peak pressure difference requirement is roughly 60–300 atm, and the peak power requirement is roughly $10^{-10}$ W. We also showed that subsequent negative pressure is sufficient to buckle the spore wall and propel the nuclei, consistent with our experimental observations. Among all the hypotheses, E-OE-PTPV-ExP is the most likely one from a physical point of view, and its schematics and predictions are summarized in *Figure 6* and the preceding paragraph. We expect new advances in dynamic ultra-fast imaging at nanoscales will experimentally test the predictions made here.

## Materials and methods

Detailed methods are provided in the Appendix. They include (1) protocols to propagate and germinated *A. algerae* spores, (2) the sample preparations, data acquisition, and analysis for SBF-SEM experiments, (3) methylcellulose experiments and the estimated effects on osmotic pressure, (4) the measurement of viscosity of germination buffers, and (5) an estimation of the osmotic pressure of *A. algerae* spores from literature.

## Acknowledgements

We thank all members of the Prakash Lab for scientific discussions and comments on figures, including Rahul Chajwa, Vishal Patil, Anesta Kothari, and Ian Ho. We thank Rebecca Konte for help and guidance on figures associated with the manuscript. We thank Joseph Sudar and Mahrukh Usmani from the Bhabha/Ekiert lab for discussion, suggestions and comments. We thank CB Cooper for advice and assistance in rheometer measurement. We thank the NYULH DART Microscopy Lab, Chris Petzold, Joseph Sall and Alice Liang for consultation and assistance with EM work. The microscopy shared resource is partially supported by the Cancer Center Support Grant P30CA016087, and Gemini300SEM with 3View was purchased with support of NIH S10 OD019974. Part of this work was performed at the Stanford Nano Shared Facilities (SNSF), supported by the National Science Foundation under award ECCS-2026822. This work was supported by Stanford University Bio-X SIGF Fellows Program (RC), Ministry of Education in Taiwan (RC), HHMI Faculty fellowship (MP), Bio-Hub Investigator Fellowship (MP), Schmidt Innovation Fellowship (MP), Moore Foundation Research Grant (MP), NSF CCC DBI1548297 (MP), NIH NIGMS R35GM128777 (DCE), Pew Charitable Trusts PEW-00033055 (GB), Searle Scholars Program SSP-2018-2737 (GB), National Institute of Allergy and Infectious Diseases

R01AI147131 (GB), Irma T Hirschl Career Scientist Award (GB), American Heart Association Postdoctoral Fellowship (PJ), Deans Undergraduate Research Fund (AD), NIH Office of Director S10OD019974 (NYU Microscopy Core).

## Additional information

### Funding

| Funder | Grant reference number | Author |
|---|---|---|
| Stanford Bio-X | SIGF Fellows Program | Ray Chang |
| Ministry of Education, Republic of China (Taiwan) | | Ray Chang |
| Howard Hughes Medical Institute | HHMI Faculty fellowship | Manu Prakash |
| Chan Zuckerberg Initiative | Bio-Hub Investigator Fellowship | Manu Prakash |
| Schmidt Futures | Schmidt Innovation Fellowship | Manu Prakash |
| Gordon and Betty Moore Foundation | Moore Foundation Research Grant | Manu Prakash |
| National Science Foundation | DBI1548297 | Manu Prakash |
| National Institute of General Medical Sciences | R35GM128777 | Damian C Ekiert |
| Pew Charitable Trusts | PEW-00033055 | Gira Bhabha |
| Searle Scholars Program | SSP-2018-2737 | Gira Bhabha |
| National Institute of Allergy and Infectious Diseases | R01AI147131 | Gira Bhabha |
| Irma T. Hirschl Trust | Career Scientist Award | Gira Bhabha |
| American Heart Association | Postdoctoral Fellowship | Pattana Jaroenlak |
| New York University | Deans Undergraduate Research Fund | Ari Davydov |

The funders had no role in study design, data collection and interpretation, or the decision to submit the work for publication.

### Author contributions

Ray Chang, Ari Davydov, Conceptualization, Data curation, Software, Formal analysis, Investigation, Visualization, Methodology, Writing – original draft, Writing – review and editing; Pattana Jaroenlak, Conceptualization, Data curation, Software, Formal analysis, Investigation, Visualization, Methodology, Writing – original draft; Breane Budaitis, Conceptualization, Data curation, Formal analysis, Validation, Investigation, Visualization, Methodology, Writing – original draft; Damian C Ekiert, Gira Bhabha, Manu Prakash, Conceptualization, Resources, Supervision, Funding acquisition, Writing – original draft, Writing – review and editing

### Author ORCIDs

Ray Chang http://orcid.org/0000-0002-9502-3306
Manu Prakash http://orcid.org/0000-0002-8046-8388

Reviewer #1 (Public Review): https://doi.org/10.7554/eLife.86638.3.sa1
Reviewer #2 (Public Review): https://doi.org/10.7554/eLife.86638.3.sa2
Author Response https://doi.org/10.7554/eLife.86638.3.sa3

# Additional files

## Supplementary files

- Supplementary file 1. Selection of potential hypotheses.
- Supplementary file 2. Summary of hypotheses.
- Supplementary file 3. Methylcellulose does not change the germination rate of *A. algerae* spores.
- Supplementary file 4. Sensitivity testing on cytoplasmic viscosity.
- Supplementary file 5. Sensitivity testing on boundary slip length ($\delta$).
- Supplementary file 6. SBF-SEM observations on spore wall buckling.
- Supplementary file 7. Sensitivity testing on cytoplasmic viscosity and boundary slip length ($\delta$), considering the 2-fold length changes in PT before and after germination.
- MDAR checklist

## Data availability

The code used in this study, including the analysis of rheometer data, and the calculation of pressure, power and total energy for each hypothesis, is available on GitHub (copy archived at *Chang, 2022*). SBF-SEM data is available in EMPIAR (EMPIAR-11367 and EMPIAR-11368). All live-cell imaging data of methylcellulose germination experiments have been deposited in an open access library in Zenodo: https://zenodo.org/record/8256725.

The following datasets were generated:

| Author(s) | Year | Dataset title | Dataset URL | Database and Identifier |
|---|---|---|---|---|
| Davydov A, Jaroenlak P, Ekiert D, Bhabha G | 2023 | SBF-SEM micrographs of A. algerae microsporidia spores, 5 min germination | https://www.ebi.ac.uk/empiar/EMPIAR-11367 | EMPIAR, EMPIAR-11367 |
| Davydov A, Jaroenlak P, Ekiert D, Bhabha G | 2023 | SBF-SEM micrographs of A. algerae microsporidia spores, 45 min germination | https://www.ebi.ac.uk/empiar/EMPIAR-11368/ | EMPIAR, EMPIAR-11368 |
| Chang R, Davydov A, Jaroenlak P, Budaitis B, Ekiert D, Bhabha G, Prakash M | 2023 | Energetics of the Microsporidian Polar Tube Invasion Machinery | https://zenodo.org/records/8256725 | Zenodo, 10.5281/zenodo.8256725 |

The following previously published dataset was used:

| Author(s) | Year | Dataset title | Dataset URL | Database and Identifier |
|---|---|---|---|---|
| Jaroenlak P, Cammer M, Davydov A, Sall J, Usmani M, Liang FX, Ekiert DC, Bhabha G | 2020 | 3-Dimensional Organization and Dynamics of the Microsporidian Polar Tube Invasion Machinery | https://zenodo.org/records/3707829 | Zenodo, 10.5281/zenodo.3707829 |

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

## Appendix 1

### A. Materials and methods

#### A.1. Propagation of *A. algerae* spores

*A. algerae* spores were propagated in Vero cells. Vero cells (ATCC CCL-81) were grown in a 25 cm² tissue culture flask using Eagle's Minimum Essential Medium (EMEM) (ATCC 30–2003) supplemented with 10% heat-inactivated fetal bovine serum (FBS) at 37 °C and with 5% CO². At 70–80% confluence, *A. algerae* (ATCC PRA-168) were added and the media was switched to EMEM supplemented with 3% FBS. Infected cells were allowed to grow for fourteen days and medium was changed every two days. To purify spores, the infected cells were detached from tissue culture flasks using a cell scraper and moved to a 15 mL conical tube, followed by centrifugation at 1300 × $g$ for 10 min at 25 °C. Cells were resuspended in 5 mL sterile distilled water and mechanically disrupted using a G-27 needle. The released spores were purified using a Percoll gradient. Equal volumes (5 mL) of spore suspension and 100% Percoll were added to a 15 mL conical tube, vortexed, and then centrifuged at 1800 × $g$ for 30 min at 25 °C. The spore pellets were washed three times with sterile distilled water and stored at 4 °C in 1 X PBS for further analyses.

### Glossary

| Term | Definition |
|---|---|
| Ungerminated spores | The entire polar tube is coiled inside the spore. |
| Incompletely germinated spores | The polar tube is partially extruded from the spore. |
| Germinated spores | The polar tube is extruded, and no polar tube remains within the spore. |
| Topological connectivity | Whether fluid flow is permitted across the end connections among organelles and sub-spaces within the spore. |
| Original polar tube content | Any material inside the polar tube prior to cargo entering the tube |
| Cargo | The content transported through the extruded polar tube; most likely the entire microsporidial cell. This content is not inside the polar tube in ungerminated spores. |
| External drag (dissipation term) | Energy dissipation between a moving polar tube and the surroundings. |
| Lubrication (dissipation term) | Energy dissipation associated with fluid flow in a thin gap. |
| Cytoplasmic flow (dissipation term) | Energy dissipation associated with fluid flow in a tube or pipe. |
| Cytoplasmic viscosity | An effective viscosity for the energy dissipation within the spore. |
| Boundary slip | An effective length scale which describes the behavior of the fluid velocity profile near a solid wall. |
| Boundary movement | The movement of the interfaces which separate different fluid compartments. |

#### A.2. Germination conditions for *A. algerae* spores

To germinate *A. algerae* spores, the following germination buffer was used: 10 mM Glycine-NaOH buffer pH 9.0 and 100 mM KCl (*Schindelin et al., 2012*). *A. algerae* spores were incubated in germination buffer at 30 °C for either 5 min or 45 min to generate two samples for SBF-SEM. The two samples were fixed in 2.5% glutaraldehyde and 2% paraformaldehyde in 0.1 M cacodylate buffer, pH 7.2 for 2 hr at room temperature. Two µL of the fixed samples were taken to observe the germination rate under the light microscope. These conditions typically yield ~70% germination.

#### A.3. Sample preparation for SBF-SEM

Fixed germinated spore samples were washed with 0.1 M sodium cacodylate buffer (pH 7.2) three times for 10 min each and post-fixed in reduced osmium (2% osmium and 1.5% potassium ferrocyanide in 0.1 M cacodylate buffer) for 1.5 hours at room temperature in the dark. Spore samples were further stained in 1% thiocarbohydrazide (TCH) solution for 20 min, followed by 2%

osmium in ddH2O for 40 min at room temperature. The sample was then embedded in 2% agar and en bloc stained with 1% uranyl acetate overnight at 4 °C in the dark, then with Walton's lead aspartate at 60 °C for 30 min. The sample was then dehydrated using a gradient of cold ethanol, and subjected to ice-cold 100% acetone for 10 min, followed by 100% acetone at room temperature for 10 min. Resin infiltration was done with 30% Durcupan in acetone for 4 hr at room temperature. The sample was kept in 50% resin in acetone at room temperature overnight, followed by 70% resin for 2 hr, 100% resin for 1 hr, and 100% resin two times for 1 hr at room temperature. The sample was then transferred to fresh 100% resin and cured at 60 °C for 72 hr, then 100 °C for 2 hr.

## A.4. SBF-SEM data collection

For SBF-SEM, the sample block was mounted on an aluminum 3View pin and electrically grounded using silver conductive epoxy (Ted Pella, catalog #16014). The entire surface of the specimen was then sputter coated with a thin layer of gold/pallidum and imaged using the Gatan OnPoint BSE detector in a Zeiss Gemini 300 VP FESEM equipped with a Gatan 3View automatic microtome. The system was set to cut 40 nm slices, imaged with gas injection setting at 40% ($2.9 \times 10^{-3}$ mBar) with Focus Charge Compensation to reduce electron accumulation charging artifacts. Images were recorded after each round of sectioning from the blockface using the SEM beam at 1.5 keV with a beam aperture size of 30 μm and a dwell time of 0.8–2.0 μs/pixel. Each frame is 22x22 μm with a pixel size of 2.2 x 2.2 × 40 nm. Data acquisition was carried out automatically using Gatan Digital Micrograph (version 3.31) software. A stack of 200–300 slices was aligned and assembled using Fiji (*Schindelin et al., 2012*). A total volume of 22 x 22 × 11 μm$^3$ was obtained from the sample block.

## A.5. SBF-SEM analysis and segmentation

Segmentation of organelles of interest, 3D reconstruction, and quantification of the spore size, volumes and PT length in the intact spores were performed using Dragonfly 4.1 software (Object Research Systems, ORS), either on a workstation or via Amazon Web Services. SBF-SEM sections were automatically aligned using the SSD (sum of squared differences) method prior to segmentation. Organelles were identified for segmentation based on color, texture, and density in the SBF-SEM 2D slices. Graphic representation of the spores and PT was performed with the Dragonfly ORS software.

Data were analyzed from both datasets that were collected: 5 min germination and 45 min germination. In addition, data from the ungerminated dataset were collected and analyzed (*Jaroenlak et al., 2020*). In total, 46 spores were segmented across all three datasets. In the 5 min germination sample, 3 ROIs were collected with approximately 80 spores in several different orientations in each ROI. Spores were randomly selected across this dataset and categorized based on germination status, including (1) ungerminated, in which the entire PT is coiled inside the spore; (2) incompletely germinated, in which the PT is partially extruded from the spore; and (3) germinated, in which the PT is extruded, and no PT remains within the spore. Of these spores, 11 incompletely germinated spores and 3 germinated spores were reconstructed in 3D to obtain volumetric and spatial information of organelles of interest. In the 45 min germination dataset, 1 ROI was collected with approximately 80 spores in several orientations. Germinated spores were randomly selected and categorized based on the presence of organelles and spore deformation ('buckling'). Of these spores, 11 germinated spores and 2 incompletely germinated spores were segmented in 3D to obtain volumetric and spatial information of organelles of interest. 50 incompletely germinated spores were also categorized based on the presence of organelles and spore deformation.

## A.6. Methylcellulose experiment

The live cell imaging of the germination process of the PT is done as previously described (*Jaroenlak et al., 2020*). In brief, 0.25 μL of purified spores of *Anncaliia algerae* were spotted on a coverslip and let the water evaporate. 2.0 μL of germination buffer (10 mM Glycine-NaOH buffer pH 9.0 and 100 mM KCl) with different concentrations (0%, 0.5%, 1%, 2%, 3%, 4%) of methylcellulose (Sigma-Aldrich catalog #M0512, approximate molecular weight 88,000 Da) was added to the slide and place the coverslip on top. The slide was imaged immediately at 37 °C on a Zeiss AxioObserver inverted microscope with a 63 x DIC objective.

Based on the molecular weight of the methylcellulose from the manufacturer and the highest concentration we used for our experiment, the additional molar concentration contributed from methylcellulose is lower than 0.45 mM, which is inconsequential compared to the existing 100 mM KCl in the germination buffer and thus should have a negligible effect on the osmotic pressure.

Also note that the germination buffer of *A. algerae* does not require hydrogen peroxide, which is a common trigger for various microsporidia species but known to oxidize polymers and change their viscosity (*Dahl et al., 1998*). Therefore for future extension of these experiments on other microsporidia species, other thickening agents must be used if the germination buffer contains hydrogen peroxide.

## A.7. Measurement of viscosity of methylcellulose solution

The viscosity of germination buffers with methylcellulose was measured using a rheometer (TA Instruments ARES-G2) at 37 °C. The temperature of the samples was equilibriated for at least 5 min before the start of the experiments. For buffers with 0%, 0.5%, and 1% methylcellulose, we used a Couette geometry (DIN Bob, 27.671 mm diameter, 41.59 mm length, SS; Cup, 29.986 mm diameter, anodized aluminum). For buffers with 2%, 3%, and 4% methylcellulose, we used a cone-and-plate geometry (40 mm diameter, 2.00° (0.035 rad) angle, 47.0 μm truncation gap, SS). A solvent well was used alongside the cone-and-plate geometry to avoid evaporation. Samples were tested in flow sweep, with the shear rate going from 1 s⁻¹ to 1000 s⁻¹, and going back from 1000 s⁻¹ to 1 s⁻¹. The viscosity at shear rate of 1000 s⁻¹ was used for the calculation, as it is closest to the estimated shear rate based on the kinematics of PT firing, except for the buffer with 0% methylcellulose, as the measurement at 1000 s⁻¹ was below the secondary flow limit of rheometer (see *Figure 3—figure supplement 3* for detail). Since the buffer with 0% methylcellulose is expected to be Newtonian fluid, we substitute the value with the viscosity measurement at shear rate of 10 s⁻¹. The surrounding viscosity measurements that we used for the theoretical calculation are 0.00067 Pa-s, 0.012 Pa-s, 0.054 Pa-s, 0.29 Pa-s, 0.71 Pa-s, and 1.16 Pa-s for buffers with 0%, 0.5%, 1%, 2%, 3%, and 4% methylcellulose, respectively.

## A.8. Estimation of osmotic pressure of *A. algerae spore*

Past experiments showed that the concentration of reducing sugar in the spores significantly increases after germination for *A. algerae* (*Undeen and Vander Meer, 1999*). According to their measurements, $10^8$ *A. algerae* spores roughly contain 400 μg sugar. Since the volume of *A. algerae* spore is 8.8 μm³, we can calculate the osmotic pressure difference (at 37 °C) generated by complete sugar conversion to be:

$$\Delta\Pi = \frac{400 \times 10^{-6}\text{g}/180\text{g/mol}}{10^8 \times 8.8 \times 10^{-15}}(0.082\text{atm-L/mol-K})(310\text{K}) = 64\text{atm} \tag{S1}$$

Note that this magnitude is comparable to the osmotic pressure needed to suppress germination in *A. algerae* spores (~60 atm) (*Undeen and Frixione, 1990*).

## A.9. Detailed explanation of energy and pressure terms for the five hypotheses

In our calculations, we start with three sources of energy dissipation – (1) external drag (energy dissipation between a moving PT and the surroundings), (2) lubrication (energy dissipation associated with fluid flow in a thin gap), and (3) cytoplasmic flow (energy dissipation associated with fluid flow in a tube or pipe) (*Figure 3—figure supplements 1–3*). In the following discussions, we defined the following symbols: $\mu_{\text{cyto}}$: cytoplasmic viscosity; $\mu_{\text{surr}}$: viscosity of the surrounding media; $v$: PT tip velocity; $L$: PT length; $L_{\text{tot}}$: total length of ejected PT; $L_{\text{sheath}}$: overlapping length of the two outermost layers of PT; $L_{\text{slip}}$: overlapping length of everted and uneverted PT; $L_{\text{open}}$: length of the PT that does not contain uneverted PT material; $D$: PT diameter; $R$: PT radius; $\epsilon$: shape factor in slender body theory, defined as $1/\ln(2L/D)$: slip length; $h_{\text{sheath}}$: lubrication thickness between the two outermost layers of PT; $h_{\text{slip}}$: lubrication thickness between everted and uneverted tube, or the cargo and everted tube; $\dot{\gamma}$: shear rate; $H$: Heaviside step function. $h_{\text{sheath}}$ was set to be 25 nm based on the observed translucent space around PT in activated spores (*Lom, 1972*), and $h_{\text{slip}}$ was set to be 6 nm based on the past images of gap thickness between PT and cargo (*Takvorian et al., 2020*). For all our calculations, we assume instantaneous development of the flow profile, which is reasonable as the flow is in low Reynolds number regime, and the time scale of PT ejection process (~1 sec) is much longer compared to the time scale required to develop the fluid profile ($\sim 10^{-4} - 10^{-8}$ sec, depending on the length scale of each profile).

### A.9.1. External drag

In the external drag term ($\mathcal{D}_{\dot{W}}$), we calculate the drag along the entire PT for Model 1 because in the jack-in-the-box mode of ejection, the entire tube is assumed to shoot out as a slender body ("a1" in *Figure 3—figure supplement 3*). The ejected portion of the PT is assumed to have drag force ($F_D$) of $2\pi\mu_{\text{surr}}vL(\epsilon + 0.806\epsilon^2 + 0.829\epsilon^3)$ according to slender body theory (*Batchelor, 1970*). The power requirement can be calculated as $F_D v$, and the pressure difference requirement calculated as $F_D/(\pi R^2)$.

For the other four hypotheses which assume a tube eversion mechanism, only the drag at the moving tip is considered since that is the only region that is moving against the surroundings ("a2" in *Figure 3—figure supplement 3*). For a spherical object with radius $R$ moving at speed $v$ in low Reynolds number regime, the drag force is $6\pi\mu_{\text{surr}}vR$. One-third of it comes from the pressure differences between the front and the back of the sphere; another one-third comes from the viscous drag on the front half, and the remaining one-third comes from the viscous drag on the other half. However, since a moving PT tip is better considered as just half of a sphere and the surrounding fluid cannot reach the back of the tip, we only consider the viscous drag on the front half. The drag force formula for Models 2–5 is thus $F_D = 2\pi\mu_{\text{surr}}vR$. Same as Model 1, the power requirement is $F_D v$, and the pressure difference requirement is $F_D/(\pi R^2)$.

Conceptually, as the drag force is linearly proportional to velocity ($v$), length scale ($l$), and surrounding viscosity ($\mu_{\text{surr}}$) in low Reynolds number regimes, and the power is the product of force and velocity, the external drag term is proportional to the square of the velocity ($\mathcal{D}_{\dot{W}} \propto \mu_{\text{surr}}v^2l$). This yields a form that is consistent with our previous descriptions.

### A.9.2. Lubrication

We next consider the energy dissipation via lubrication ($\mathcal{L}_{\dot{W}}$). We assume all the lubrication processes to be the sliding Couette flow between two concentric cylinders, with generalization to include the effect of boundary slip. We did not use actual lubrication theory for calculation as the exact gap height profile during the PT ejection process is not known, and our calculation can thus be considered as a lower bound estimation. We also did not account for the changes in flow profile at the end corner of each space, as those secondary fluid flow only spans a length scale comparable to the thickness of the gap (~10 nm) and much smaller compared to the length scale of PT (μm-scale).

For this flow profile with one boundary at velocity $v$ and the other boundary at velocity 0, with gap height $h$, boundary slip length $\delta$, and overlapping length $\ell$, the fluid shear rate is homogeneous: $\dot{\gamma} = \frac{v}{h + 2\delta}$. The dissipation power is in the form of $\mathcal{L}_{\dot{W}} = \pi\mu_{\text{cyto}}\left(\frac{v}{h + 2\delta}\right)^2 \ell(2Rh + h^2)$, proportional to the square of shear rate ($\dot{\gamma}^2 \propto (v/(h + 2\delta))^2$) times the volume of the gap zone ($\pi\ell(2Rh + h^2)$). The total lubrication drag associated is proportional to the fluid stress at the boundary ($\mu_{\text{cyto}}\frac{v}{h + 2\delta}$) times the surface area ($2\pi R\ell$). We thus estimate the pressure difference requirement associated with lubrication as $(\mu_{\text{cyto}}\frac{v}{h + 2\delta})(2\pi R\ell)/(\pi R^2) = \frac{2\mu_{\text{cyto}}v\ell}{R(h + 2\delta)}$.

As for the exact terms we considered, we first account for lubrication in the PT pre-germination. Cross-sections from previous TEM studies have shown that the PT is likely composed of concentric layers (*Xu and Weiss, 2005*), and the translucent space between the two outermost layers enlarges before PT ejection. We thus account for lubrication between the two outermost layers ('b1' in *Figure 3—figure supplement 3*). As this space is visible before the PT ejects, it should be accounted in all five hypotheses. For Model 1, the overlapping length of this space should be $\frac{1}{2}(L_{\text{tot}} - L(t))$, as this topology naturally predicts a two-times difference in PT length before and after germination. For Model 2 - Model 5, the overlapping length of this space is $(L_{\text{tot}} - 2L(t))H(L_{\text{tot}} - 2L(t))$. It has this form with Heaviside step function because the overlapping space between the two outermost layers would disappear when the eversion is halfway through.

Second, we include the lubrication between the uneverted part of the tube (blue) and the everted tube (green) for Model 2 - Model 5 ('b2' in *Figure 3—figure supplement 3*). The height of this overlapping space is assumed to be the same as the gap height between cargo and PT. The length of this overlapping segment is calculated as $\min(L(t), L_{\text{tot}} - L(t))$, where min selects the minimum of the two terms. Before the eversion is halfway through, the overlapping length between uneverted and everted PT is simply $L(t)$, while after the eversion is halfway through, the overlapping length becomes $L_{\text{tot}} - L(t)$.

Finally, for Model 5, we also consider the lubrication between cargo and everted PT ('b3' in *Figure 3—figure supplement 3*). We consider this because this hypothesis requires both the original PT space and posterior vacuole to be open to the external environment but not to the sporoplasm, and this topology requires the cargo to be separated from the PT by a fluid gap that is connected to the fluid in the external environment. The overlapping length of this is calculated as $(2L(t) - L_{tot})H(2L(t) - L_{tot})$. It has this form with Heaviside step function because the cargo can only enter PT after the eversion is halfway through.

## A.9.3. Cytoplasmic flow

In the cytoplasmic flow term ($\mathcal{C}_{\dot{W}}$), the dissipation power also scales to the square of shear rate times the volume of dissipative fluid. We assume the fluid flow to be Poiseuille flow with generalization to include the effect of boundary slip, spanning length of $\ell$. In a cylindrical coordinate with axial direction as $z$ and radial direction as $r$, the velocity profile of Poiseuille flow with boundary slip would be $u_z(r) = \frac{1}{4\mu_{cyto}}((R+\delta)^2 - r^2)(-\frac{dp}{dz})$, where $u_z(r)$ is the fluid velocity in $z$ direction at radial position $r$, and $(-\frac{dp}{dz})$ is the pressure gradient. The negative sign comes from the fact that the fluid flow is in the opposite direction to the pressure gradient, as fluid flows from high pressure to low pressure. The volumetric flow rate ($Q$) can thus be derived as $Q = \int_0^R u_z 2\pi r dr = \frac{\pi}{2\mu_{cyto}}(-\frac{dp}{dz})(\frac{1}{2}(R+\delta)^2 R^2 - \frac{1}{4}R^4)$. The mean velocity calculated from volumetric flow rate is thus $\bar{u}_z = Q/(\pi R^2)$ and set to be the same as the fluid velocity of the space $v$. The required pressure differences can thus be written as $\Delta p = \frac{2\mu_{cyto}\ell v}{(\frac{1}{2}(R+\delta)^2 - \frac{1}{4}R^2)}$. From the velocity profile, we can derive the fluid shear rate at radial position $r$ to be $\dot{\gamma}(r) = \frac{du_z}{dr} = -\frac{r}{2\mu_{cyto}}(-\frac{dp}{dz})$. The total power required can be calculated as the volume integral of $\mu_{cyto}\dot{\gamma}^2$: $\mathcal{C}_{\dot{W}} = \ell \int_0^R (2\pi r)\mu_{cyto}\dot{\gamma}^2 dr = \frac{\pi}{2}\mu_{cyto}\ell v^2 \frac{R^4}{(\frac{1}{2}(R+\delta)^2 - \frac{1}{4}R^2)^2}$.

For Model 1, as the entire PT with the internal fluid is moving, we assume the fluid flow inside PT has a homogeneous velocity. This homogeneous velocity profile excludes the shear dissipation in power calculation, but still requires the pressure field to keep up the velocity (otherwise the velocity profile will have a high velocity at the wall but low velocity near the center) ('c1' in *Figure 3—figure supplement 3*). For Model 3, as the cytoplasm, original PT content, and external environment are connected, the boundary movement of PT eversion will only drive fluid flow in the lubrication thin spaces and much less within the PT. For Models 2, 4, and 5, the fluid flow of cytoplasm within the PT after the eversion is halfway through is considered ('c1' in *Figure 3—figure supplement 3*). For Models 4 and 5, the additional flow to extrude the original PT content into the external environment is also considered ('c2' in *Figure 3—figure supplement 3*).

Combining all the aforementioned calculation, for each observed spore germination event, we can compute the peak power requirement, peak pressure difference requirement, and total energy requirement of the PT firing process for each hypothesis, according to the equations we formulated in *Figure 3—figure supplements 1 and 2*.

## A.9.4. Adaptation to account for PT length changes

In the main text, we did not account for the twofold length changes of PT before and after germination for Model 2–5 (Model 1 naturally accounts for the twofold length changes of PT). If we want to include that effect, the formula for $L_{sheath}$, $L_{open}$ and $L_{slip}$ for Models 2–5 have to be modified as follow:

$$
\begin{aligned}
L_{sheath}(t) &= \left(\frac{L_{tot}}{\lambda} - \left(1 + \frac{1}{\lambda}\right)L(t)\right)H\left(\frac{L_{tot}}{\lambda} - \left(1 + \frac{1}{\lambda}\right)L(t)\right) \\
L_{open}(t) &= \left(\left(1 + \frac{1}{\lambda}\right)L(t) - \frac{L_{tot}}{\lambda}\right)H\left(\left(1 + \frac{1}{\lambda}\right)L(t) - \frac{L_{tot}}{\lambda}\right) \\
L_{slip}(t) &= \min\left(L(t), \frac{L_{tot} - L(t)}{\lambda}\right)
\end{aligned}
\tag{S2}
$$

, where $\lambda$ is the fold-changes in PT length. We have reported the results in *Supplementary file 7* (for $\lambda = 2$), and the overall ranking among the proposed hypotheses does not change.

