## [Editor Report · eLife assessment]

This **important** study combines experiments and fluid mechanics modeling to determine the mechanism of the ultrafast ejection of the polar tube of the Microsporidia parasite and of transport through this tube. The methods and the analysis, based on the variation of the viscosity of the external medium, are **compelling** and allow for the first time to discriminate among proposed ejection mechanisms. This approach where simple physical principles are used for distinguishing between mechanisms when the precise geometry is inaccessible through imaging is potentially applicable to other systems in microbiology.

---

## [Referee Report · Reviewer #1 (Public Review)]

The authors used mathematical models to explore the mechanism(s) underlying the process of polar tube extrusion and the transport of the sporoplasm and nucleus through this structure. They combined this with experimental observations of the structure of the tube during extrusion using serial block face EM providing 3 dimensional data on this process. They also examined the effect of hyperosmolar media on this process to evaluate which model fit the predicted observed behavior of the polar tube in these various media solutions. Overall, this work resulted in the authors arriving at a model of this process that fit the data (model 5, E-OE-PTPV-ExP). This model is consistent with other data in the literature and provides support for the concept that the polar tube functions by eversion (unfolding like a finger of a glove) and that the expanding polar vacuole is part of this process. Finally, the authors provide important new insights into the bucking of the spore wall (and possible cavitation) as providing force for the nucleus to be transported via the polar tube. This is an important observation that has not been in previous models of this process.

---

## [Referee Report · Reviewer #2 (Public Review)]

The paper follows a recent study by the same team (Jaroenlak et al Plos Pathogens 2020), which documented the dramatic ejection dynamics of the polar tube (PT) in microsporidia using live-imaging and scanning electron microscopy. Although several key observations were reported in this paper (the 3D architecture of the PT within the spore, the speed and extent of the ejection process, the translocation dynamics of the nucleus during germination), the precise geometry of the PT during ejection remain inaccessible to imaging, making it difficult to physically understand the phenomenon.

This paper aims to fill this gap with an indirect "data-driven" approach. By modeling the hydrodynamic dissipation for different unfolding mechanisms identified in the literature and by comparing the predictions with experiments of ejection in media of various viscosities, authors shows that data are compatible with an eversion (caterpillar-like) mechanism but not compatible with a "jack-in-the-box" scenario. In addition, the authors observe that most germinated spores exhibit an inward bulge, which they attribute to buckling due to negative pressure difference. They suggest that this buckling may be a mean of pushing the nucleus out of the PT during the final stage of ejection.

Major strengths:

The most compelling aspect of the study is the experimental analysis of the ejection dynamics (velocity, ejection length) in medium of various viscosities over 3 orders of magnitudes, which, combined with a modeling of the viscous drag of the PT tube, provides very convincing evidence that the unfolding geometry is not a global displacement of the tube but rather an apical extension, where the motion is localized at the end of the tube.

The systematic classification of the different unfolding scenarios, consistent with the previous literature, and their confrontation with data in terms of energy, pressure and velocity also constitute an original approach in microbiology, where in-situ and real time geometry is often difficult to access.

Major weaknesses:

The revised version has clarified some details of the model, adding a paragraph and a figure in the Sup Mat. However, in my opinion, it remains difficult to understand the precise topology and ejection mechanism from the various sketches presented in the article.

The article does not address the mechanical driver (force) of ejection, and the role of pressure is unclear. The revised version replaced the term "negative pressure" with "negative pressure difference", arguing that a positive or negative pressure difference could not be differentiated. However, it is not clear how a lower pressure in the spore than in the bath could eject the tube outside.

---

## [Author Response]

The following is the authors’ response to the original reviews.

Thank you for the response and reviews of our manuscript eLife-RP-RA-2023-86638 “Energetics of the Microsporidian Polar Tube Invasion Machinery”. We are grateful for the comments and constructive criticism from all three reviewers, which have helped us to improve our manuscript.

As a summary to the editor, we here provide a list of the major revisions we have implemented to address all the comments provided by the referees.

1. We added Supplementary Section A.9 and Figure S4 to explain the details of calculation and have magnified sketches of flow fields.

2. We clarified the term "required pressure" to "required pressure differences", and explained that the same pressure differences can be achieved by either positive or negative pressure. We invoke the fact that the spore wall buckled inward to deduce that germination is a negative pressure process.

1. We only rank the hypotheses based on calculation of total energy requirement. The peak pressure and peak power requirement calculations are now just for quantitative reference. The ranking of hypotheses does not change.

2. We clarified the definition of topological connections in Section "Systematic evaluation of possible topological configurations of a spore," making it explicit that the topological questions listed only involved the "original PT content" (not PT space at all time).

Thank you again for the opportunity to revise our work. We attach a point-by-point response to the referees below.

**Public Reviews:**

**Reviewer #1 (Public Review):**
1. The authors used mathematical models to explore the mechanism(s) underlying the process of polar tube extrusion and the transport of the sporoplasm and nucleus through this structure. They combined this with experimental observations of the structure of the tube during extrusion using serial block face EM providing 3 dimensional data on this process. They also examined the effect of hyperosmolar media on this process to evaluate which model fit the predicted observed behavior of the polar tube in these various media solutions.

We thank the reviewer for their accurate summary of our work. One subtle point, however, is that we examine the effect of hyperviscous media on the polar tube extrusion process, rather than hyperosmolar media. In Supplementary Section A.6 of our updated manuscript, we have shown that the changes in osmolarity due to methylcellulose is negligible.

1. Overall, this work resulted in the authors arriving at a model of this process that fit the data (model 5, E-OE-PTPV-ExP). This model is consistent with other data in the literature and provides support for the concept that the polar tube functions by eversion (unfolding like a finger of a glove) and that the expanding polar vacuole is part of this process. Finally, the authors provide important new insights into the buckling of the spore wall (and possible cavitation) as providing force for the nucleus to be transported via the polar tube. This is an important observation that has not been in previous models of this process.

We thank the reviewer for acknowledging the novelty and importance of our study.

**Reviewer #2 (Public Review):**
1. Microsporidia has a special invasion mechanism, which the polar tube (PT) ejects from mature spores at ultra-fast speeds, to penetrate the host and transfer the cargo to host. This work generated models for the physical basis of polar tube firing and cargo transport through the polar tube. They also use a combination of experiments and theory to elucidate possible biophysical mechanisms of microsporidia. Moreover, their approach also provided the potential applications of such biophysical approaches to other cellular architecture.

We thank the reviewer for their accurate summary and acknowledging the potential applications on other organisms.

1. The conclusions of this paper are mostly well supported by data, but some analyses need to be clarified. According to the model 5 (E-OE-PTPV-ExP) in P42 Fig. 6, is the posterior vacuole connected with the polar tube? If yes, how does the nucleus unconnected with the posterior vacuole enter the polar tube?

As we mentioned in our glossary and detailed in Section "Systematic evaluation of possible topological configurations of a spore", Model 5 requires the "original PT content" (any material inside the PT prior to cargo entering the tube) to permit fluid flow to posterior vacuole and external environment post anchoring disc rupture, but cannot permit fluid flow to the sporoplasm that is transported through the tube. As the the germination process progresses, our model does not require the connection between PT and posterior vacuole to be maintained afterwards, and that creates space allowing sporoplasm (including nucleus) sporoplasm (including nucleus) to enter PT space through fluid entrainment. We have clarified the definitions in Section "Systematic evaluation of possible topological configurations of a spore" and have additional clarification in the caption of Fig. 6 in the updated manuscript.

1. In Fig. 6, would the posterior vacuole become two parts after spore germination? One part is transported via the polar tube, and the other is still in the spore. I recommend this process requires more experiments to prove.

According to our Model 5, the membrane connection between PT and posterior vacuole must be broken for the infectious cargo to extrude. However, our current data does not allow us to prove nor disprove the membrane fission event. In theory, the membrane content in PT can potentially be severed into multiple parts by Plateau-Rayleigh instability, an interfacial-tension-driven fluid thread breakup mechanism. Note that it is possible to have membrane fission at the time scale of germination process, as when the time scale of shearing is faster than the viscoelastic time of lipid membranes (roughly 10 msec), membrane fission can happen (Morlot & Roux 2013). For time scale longer than viscoelastic time of lipid membrane, protein complexes like dynamin would be required for membrane fission. Future cryo-EM study of the vacuole-PT connection at the anterior tip (and in the spore as a whole) is needed to clarify the physical process. We added this discussion in Section "Predictions and proposed future experiments".

**Reviewer #3 (Public Review):**
Abstract:The paper follows a recent study by the same team (Jaroenlak et al Plos Pathogens 2020), which documented the dramatic ejection dynamics of the polar tube (PT) in microsporidia using live-imaging and scanning electron microscopy. Although several key observations were reported in this paper (the 3D architecture of the PT within the spore, the speed and extent of the ejection process, the translocation dynamics of the nucleus during germination), the precise geometry of the PT during ejection remain inaccessible to imaging, making it difficult to physically understand the phenomenon.This paper aims to fill this gap with an indirect "data-driven" approach. By modeling the hydrodynamic dissipation for different unfolding mechanisms identified in the literature and by comparing the predictions with experiments of ejection in media of various viscosities, authors shows that data are compatible with an eversion (caterpillar-like) mechanism but not compatible with a "jack-in-the-box" scenario. In addition, the authors observe that most germinated spores exhibit an inward bulge, which they attribute to buckling due to internal negative pressure and which they suggest may be a mean of pushing the nucleus out of the PT during the final stage of ejection.

We thank the reviewer for their accurate summary of our work.

Major strengths:Probably the most impressive aspect of the study is the experimental analysis of the ejection dynamics (velocity, ejection length) in medium of various viscosities over 3 orders of magnitudes, which, combined with a modeling of the viscous drag of the PT tube, provides very convincing evidence that the unfolding mechanism is not a global displacement of the tube but rather an apical extension mechanism, where the motion is localized at the end of the tube. The systematic classification of the different unfolding scenarios, consistent with the previous literature, and their confrontation with data in terms of energy, pressure and velocity also constitute an original approach in microbiology where in-situ and real time geometry is often difficult to access.

We thank the reviewer for acknowledging the novelty and importance of our study.

Major weaknesses:1a. While the experimental part of the paper is clear, I had (and still have) a hard time understanding the modeling part. Overall, the different unfolding mechanisms should be much better explained, with much more informative sketches to justify the dissipation and pressure terms, magnifying the different areas where dissipation occurs, showing the velocity field and pressure field, etc.

We thank the reviewer for their comments and suggestions. In the Figure S4 and SI Section A.9 of the updated manuscript, we have magnified the sketches with flow field, and added a detailed explanation of the derivations of dissipation terms.

1b. In particular, a key parameter of eversion models is the geometry of the lubrication layers inside and outside the spore (h_sheath, h_slip). Where do the values of h_sheath and h_slip come from? What is the physical process that selects these parameters?

As we described in SI Section A.9, h_sheath was set to be 25 nm based on the observed translucent space around PT in activated spores (Lom 1972), and h_slip was set to be 6 nm based on the observed gap thickness between PT and cargo (Takovarian et al. 2020). Although we don't expect these numbers to be the same for each spore, the uncertainty in these two parameters are much less than the uncertainty in cytoplasmic viscosity (which varies several orders of magnitude) and boundary slip length. Our sensitivity testing on cytoplasmic viscosity and boundary slip length thus covers any uncertainty in h_sheath or h_slip already.

1c. For clarity, the figures showing the unfolding mechanics in the different scenarios should be in the main text, not in the supplemental materials.

We have added Figure S4 and SI Section A.9 to explain the details of our sketches. We believe, however, putting all the details of the mechanics and how each term is derived in the main text may detract from the flow of the manuscript, and result in it being less accessible to readers who are not as familiar with the physics. We therefore decided to keep this information in supplemental materials.

2a. The authors compute and discuss in several places "the pressure" required for ejection, but no pressure is indicated in the various sketches and no general "ejection mechanism" involving this pressure is mentioned in the paper.

In the updated manuscript, we have changed the term “pressure” to “pressure difference” or “required pressure difference”. We did not calculate the detailed pressure field around each structure, but only estimated the required pressure difference to overcome the drag force and drive fluid flow in various spaces. We also clarified this point in Section "Developing a mathematical model for PT energetics".

Also, as we mentioned in Section “Posterior vacuole expansion and the role of osmotic pressure”, we made no assumptions on how the pressure difference is generated in this paper. The unfolding mechanism of polar tube, how eversion is sustained, and the driving mechanism are ongoing research projects, and we decided not to make premature comments on that without strong support from experiments or simulation results.

2b. What is this "required pressure" and to what element does it apply?

The “required pressure” in the manuscript indicates the required pressure difference between the spore and the tip of the polar tube for it to push the tip forward and sustain the fluid flow within the polar tube. In the updated manuscript, we thus changed the term “required pressure” to “required pressure difference”. We also added this clarification to Section "Developing a mathematical model for PT energetics".

2c. I understand that the article focuses on the dissipation required to the deployment of the PT but I find it difficult to discuss the unfolding mechanism without having any idea on the driving mechanism of the movement. How could eversion be initiated and sustained?

As we mentioned in Section “Posterior vacuole expansion and the role of osmotic pressure”, we made no assumptions on how the energy, pressure or power is generated in this paper. We agree that the unfolding mechanism of the polar tube, how eversion is sustained, and the driving mechanism are important questions, and these are ongoing research projects. As no assumptions about this are required for our models, we decided not to comment on these aspects without strong support from experiments or simulation results. We have clarified this in Section “Posterior vacuole expansion and the role of osmotic pressure” of the updated manuscript.

1. Finally, the authors do not explain how pressure, which appears to be a positive, driving quantity at the beginning of the process, can become negative to induce buckling at the end of ejection. Although the hypothesis of rapid translocation induced by buckling is interesting, a much better mechanistic description of the process is needed to support it.

As discussed in Point 2-b above, the “required pressure” actually means “required pressure difference”. The same pressure difference can possibly be achieved by either positive pressure (the spore has a higher pressure than the ambient, pushing the fluid into PT) or negative pressure (the PT tip has a lower pressure than the ambient, sucking the fluid from the spore). Hydrodynamic dissipation analysis alone cannot tell the differences between positive or negative pressure, as it only tells you the required pressure differences between the spore and the polar tube tip. It will have to be inferred from the implied mechanisms or other evidence. We added these discussions in the 4th paragraph of Section "Developing a mathematical model for PT energetics" in the updated manuscript.

That being said, from our observations of buckled spore walls, it is still sufficient to deduce that the polar tube ejection process is a negative pressure driven process. For the spore wall to buckle inwards, the ambient pressure has to be higher than the pressure within the spore, but that would contradict with the positive pressure hypothesis as elaborated above. We added these clarifications in the 2nd paragraph of Section "Models for the driving force behind cargo expulsion".

References:

Lom, J. (1972). On the structure of the extruded microsporidian polar filament. Zeitschrift Für Parasitenkunde, 38(3), 200–213.

Takvorian, P. M., Han, B., Cali, A., Rice, W. J., Gunther, L., Macaluso, F., & Weiss, L. M. (2020). An Ultrastructural Study of the Extruded Polar Tube of Anncaliia algerae (Microsporidia). The Journal of Eukaryotic Microbiology, 67(1), 28–44.

Morlot, S., & Roux, A. (2013). Mechanics of dynamin-mediated membrane fission. Annual Review of Biophysics, 42, 629–649.

**Reviewer #1 (Recommendations For The Authors):**
The work is solid and supported by the experimental data presented, the literature and the biophysical modeling.1. The model (Model 5) indicates that the polar tube is connected to the posterior vacuole and that the contents of this vacuole may be transported by the polar tube before the sporoplasm. This needs experimental validation in the future, which will require the identification of posterior vacuole markers (i.e. proteins specific to this structure). I find the topology of this idea difficult to understand. If the polar tube is outside of the sporoplasm membrane then how does it connect to the posterior vacuole? If the expanded posterior vacuole is still in the spore at the end of germination then how does the sporoplasm get out?

Model 5 requires the "original PT content" (any material inside the PT prior to cargo entering the tube) to permit fluid flow to posterior vacuole and external environment post anchoring disc rupture, but cannot permit fluid flow to sporoplasm. As the germination process progresses, our model does not require the connection between PT and posterior vacuole to be maintained afterwards, and that creates space allowing sporoplasm (including nucleus) to enter PT space through fluid entrainment.

We agree with the reviewer that the specific predictions from Model 5 need to be experimentally validated in the future, and identification of posterior vacuole markers is a good direction. We have mentioned this in Section "Predictions and proposed future experiments".

1. I have always thought that the polaroplast was the initial cargo in the polar tube and that this formed the limiting membrane of the sporoplasm and nucleus after passage through the polar tube (i.e., the limiting membrane of the sporont).

In this manuscript, we only analyze the possible topology of the organelles that are relevant for energy dissipation calculations. Our final hypothesis (E-OE-PTPV-ExP) indicates that there is a limiting membrane of the infectious cargo as they pass through PT, but the energy calculation cannot tell you where this membrane comes from. That being said, our final hypothesis is consistent with the common belief that polaroplast provides the limiting membrane of the sporoplasm, even though our analysis neither proved nor disproved it.

1. I understand that the model indicates that during eversion the end of the PT moves away from the posterior vacuole allowing the sporoplasm access to the PT lumen, however, I am not clear how this process occurs (although I understand the reason that this model was the best fit for the available data). Does the model distinguish between connected (as in the PV is in the polar tube lumen) to the idea of it being in proximity (i.e. the PT is at the PV at the start of eversion)?

As we mentioned in our reply to Point 1 of the same reviewer above, "connectivity" simply means whether fluid flow is permitted across the end connections among organelles and sub-spaces within the spores. For Model 5, the content of posterior vacuole can pass to the original PT content and to the external environment post anchoring disc disruption through fluid flow, but not to sporoplasm. However, as the germination progresses, the PT does not have to maintain its spatial proximity or membrane connection to posterior vacuole, as the topological connectivity questions are pertaining to the "original PT content". We clarified this point in Section "Systematic evaluation of possible topological configurations of a spore" in the updated manuscript.

**Reviewer #2 (Recommendations For The Authors):**
1. The connection of polar tube and posterior vacuole need to be analyzed by Cryo -EM.

We thank the reviewer for their comments. This work is underway.

**Reviewer #3 (Recommendations For The Authors):**
1a. As stated in the public review, the explanation and description of the unfolding mechanism should be much better described and associated with clear sketches, magnifying all the areas where the flow shear rate is concentrated (surrounding zone, lubrication inside and outside the spore, etc) and drawing the velocity field, the boundary solid motion and pressure distribution in order to clearly understand, for each model, the dissipation and pressure terms given in figs. S2 and S3.

In the updated manuscript, we added Figure S4 to enlarge all the regions where fluid shear is considered, with sketches of velocity fields.

1b. This is particularly important for explaining the eversion models (see comment in the Public Review) but even the "jack-in-the-box" model sketched in Fig. S2 is confusing: Why does the blue tube disappear outside the spore? What happens to the tube in this case?

The blue tube in the sketch of Model 1 in Fig. S2 is the fluid between the two outermost layers of PT, not the PT itself. We have clarified that in the newly added Fig. S4.

1. Many ejection mechanisms based on the deployment of invaginated appendages have been described in the literature (e.g. Zuckerkandl Biol. Bull. 1950, Karabulut et al Nat. Com. 2022) and also mimicked for robotic applications (e.g. Hawkes et al Science Robotics 2017). Although this is not the main topic of the paper, it would be very useful if the authors could discuss in the introduction the most acceptable theory for motion generation (eversion driven by an overpressure in the spore?). In the current version, this comes too late in the discussion.

As we discussed in Section “Lack of biophysical models explaining the microsporidian infection process”, PT eversion is the most widely accepted hypothesis because of experimental evidence (e.g. microscopic observations of PT extrusions, and pulse-labeling of half-ejected tubes). However, whether or not it is driven by an overpressure in the spore remains controversial. In fact, our observations of inwardly buckled spores indicates that the ejection process likely involves negative pressure.

In our work, we thus take a data-driven approach to generate models for the physical basis of PT extrusion process, without immediately assuming that eversion is the correct hypothesis. It would therefore not make sense to have elaborated discussion on other eversion mechanisms in Introduction.

1. About the physical constraints, I understand that the stored energy must be the same when the viscosity is changed (by conservation of energy), but what physical basis do you have for requiring that the power and pressure also be the same (lines 295-298)? For e.g. when a spring is stretched and released in a very viscous fluid without inertia, the total energy dissipated is the same whatever the viscosity but the power is not the same. The formulation of the chosen physical constraints should be better justified.

We thank the reviewer for their feedback. In our updated manuscript, we only use total energy requirement for the ranking, and the peak pressure difference requirement and peak power requirements are calculated just for quantitative reference. The ranking of the 5 hypotheses does not change.

1. About the mechanism for cargo translocation, authors should explain the physical origin of the hypothetical negative pressure. How could the initial positive pressure become negative?

As we mentioned in our reply to Point 3 of the same reviewer in the public review, the “required pressure” actually means “required pressure difference”. The same pressure difference can possibly be achieved by either positive pressure (the spore has a higher pressure than the ambient, pushing the fluid into PT) or negative pressure (the PT tip has a lower pressure than the ambient, sucking the fluid from the spore). Hydrodynamic dissipation analysis alone cannot tell the differences between positive or negative pressure, as it only tells you the required pressure differences between the spore and the polar tube tip. It will have to be inferred from the implied mechanisms or other evidence. We added these discussions in the 4th paragraph of Section "Developing a mathematical model for PT energetics" in the updated manuscript.

That being said, from our observations of buckled spore walls, it is still sufficient to deduce that the polar tube ejection process is a negative pressure driven process. For the spore wall to buckle inwards, the ambient pressure has to be higher than the pressure within the spore, but that would contradict with the positive pressure hypothesis as elaborated above. We added these clarifications in the 2nd paragraph of Section "Models for the driving force behind cargo expulsion".

More minor comments:1. The videos are amazing but it is not clear if the PT is ejected through a bulk fluid or if the spores (and ejected PT) are in contact with a solid.

As described in Supplementary Section A.6, purified spores were spotted on a coverslip and let water evaporate. 2.0 μL of germination buffer (10 mM Glycine-NaOH buffer pH 9.0 and 100 mM KCl) with different concentration (0%, 0.5%, 1%, 2%, 3%, 4%) of methylcellulose was added to the slide and place the coverslip on top. So the spore is attached to the coverslip and ejected through a bulk liquid of germination buffer.

1. S2 caption: please be precise that H is the Heaviside step function.

We have updated the captions for both Figure S2 and S3 to make it explicit.

1. Line 233 a pi is missing, no?

We thank the reviewer for their careful read. We have corrected that.

1. The notations are quite unfortunate and confusing. In fluid mechanics capital D usually refers to the dissipation, capital C to the drag coefficient. It would be much clearer to call D the dissipation power (in Watt) and P the pressure requirement (in Pa), whatever the mechanism and put the different contribution (drag, lubrication, cytoplasm flow) in subscript.

We thank the reviewer for their feedback. The notation of this paper is challenging as there are many symbols while keeping everything relatively intuitive to both people with biology background and physics background. We will keep these feedback in mind in our future work.

1. Fig S2: what is D (in the formula of the total dissipation power)? Why not use R instead?

D is the PT diameter, as we mentioned in the caption. We keep that as it is used in the definition of the shape factor.

1. Fig S3 why the pressure requirement for the "jack-in-the-box" hypothesis is 2\mu (v*L*f(epsilon)/R^2)?

We have now elaborated the calculation in SI Section A.9.

1. Lines 486-497: Although shear thinning fluids have their viscosity that decreases with the shear rate, in most cases the resistance (stress) still increases with speed with these fluids. Is mucin a "velocity-weakening" fluid, i.e. a fluid in which stress decreases when shear rate increases.

We agree that stress still increases with speed for most shear thinning fluids. The mechanical properties of mucin solution strongly depend on its compositions and buffers. In our discussion, we thus simply mention this possibility without claiming whether mucin (or other biopolymer environment that microsporidia species actually experience in vivo) is a velocity-weakening fluid or not.